# Bayesian Learning of Adaptive Koopman Operator with Application to Robust Motion Planning for Autonomous Trucks

## Abstract

Koopman theory has recently been shown to enable an efficient data-driven approach for modeling physical systems, offering a linear framework despite underlying nonlinear dynamics. It is, however, not clear how to account for uncertainty or temporal distributional shifts within this framework, both commonly encountered in real-world autonomous driving with changing weather conditions and time-varying vehicle dynamics. In this work, we introduce BLAK, **B**ayesian **L**earning of **A**daptive **K**oopman operator to address these limitations. Specifically, we propose a Bayesian Koopman operator that incorporates uncertainty quantification, enabling more robust predictions. To tackle distributional shifts, we propose an online adaptation mechanism, ensuring the operator remains responsive to changes in system dynamics. Additionally, we apply the architecture to motion planning and show that it gives fast and precise predictions. By leveraging uncertainty awareness and real-time updates, our planner generates dynamically accurate trajectories and makes more informed decisions. We evaluate our method on real-world truck dynamics data under varying weather conditions—such as wet roads, snow, and ice—where uncertainty and dynamic shifts are prominent, as well as in other simulated environments. The results demonstrate our method's ability to deliver accurate, uncertainty-aware open-loop predictions for dynamic systems.

## 1 Introduction

In the context of autonomous driving, a key challenge is ensuring reliable vehicle performance in various weather conditions. Autonomous vehicles must navigate environments such as icy, wet, or snowy roads. These conditions significantly alter vehicle behavior, making the driving task more difficult and increasing the likelihood of safety-critical incidents. Reduced traction impacts the vehicle's ability to maintain grip, potentially resulting in skidding or sliding. Furthermore, steering inputs become less reliable, often resulting in understeer, where the vehicle's actual turning radius exceeds the intended path, or makes it even infeasible to follow that path, complicating trajectory tracking (Russell & Gerdes, 2014; Berntorp et al., 2020). On inclines, low traction makes climbing hills hazardous, while descending can be even more treacherous, especially for heavy vehicles like trucks, which face the added danger of jackknifing or trailer swings and instability. The constantly shifting environmental factors lead to what is known as a distributional shift, where the underlying conditions change in ways that were not anticipated during the initial modeling process. To add to the complexity, vehicles with varying load distributions exhibit different inertia profiles, which influence how the system behaves under different driving conditions. This makes it essential for models to be adaptable, capable of accurately reflecting these changes in real time.

In severe weather conditions, autonomous motion planning must account for the significant impact of environmental factors on vehicle behavior. On low-friction surfaces like icy or wet roads, a vehicle may not behave as expected, rendering previously planned trajectories unsafe or entirely unfeasible. For instance, extended braking distances are required to prevent accidents in low-traction scenarios, while steering adjustments may be insufficient to maintain the intended path. These deviations from normal vehicle behavior increase the complexity of navigating hazardous conditions, underscoring the need for motion planning that goes beyond basic geometric or kinematic models. To address

these challenges, motion planning must be dynamically aware of the vehicle's physical limitations and the real-time conditions it faces (Hu et al., 2022; Svensson et al., 2021). This requires integrating models that can adapt to changing weather conditions, load distributions, and vehicle inertia (Svensson & Törngren, 2021).

Recently, learning-based approaches for modeling physical systems have emerged in a variety of domains, such as robotics (Han et al., 2021; Shi & Meng, 2022), autonomous driving (Xiao et al., 2024; Wang et al., 2024), and general system identification of dynamical systems (Atuonwu et al., 2010; Nerrand et al., 1994; Baruch & Mariaca-Gaspar, 2009; Akpan & Hassapis, 2011). Among these, the Koopman operator has emerged as a powerful data-driven tool for modeling nonlinear systems with unknown dynamics (Koopman, 1931). By mapping nonlinear dynamics into a linear framework through the propagation of observables in an embedded space, the Koopman operator facilitates the application of linear analysis and control techniques (Mauroy et al., 2020; Bevanda et al., 2021). However, significant challenges remain in data-driven Koopman modeling. These challenges are particularly evident when addressing rigorous uncertainty quantification, encompassing both aleatoric and epistemic uncertainties, as well as managing distribution shifts in time-varying dynamics. Such challenges become particularly evident in dynamic environments, like varying road conditions, where Koopman-based models often struggle with real-time adaptation and effective uncertainty quantification. These limitations pose critical obstacles to ensuring the safety and robustness of systems operating in unpredictable and evolving settings.

This work presents a novel framework that utilizes a stochastic Koopman operator to address the challenges identified earlier. By placing a probability distribution over the Koopman operator, the proposed approach explicitly models uncertainties in system dynamics, including both state transitions (model uncertainty) and observation noise (data uncertainty). Additionally, this probabilistic formulation enables the model to adapt to distribution shifts, ensuring robust performance in time-varying dynamic environments. Contrasting with traditional methods that estimate a single deterministic Koopman operator, our approach learns a distribution over an infinite set of operators, each weighted by its probability of explaining the observed data. This ensemble methodology incorporates multiple hypotheses about the system's dynamics, enhancing adaptability and predictive accuracy as new data becomes available.

The organization of this paper is as follows: in section 2 we go through the background and related works. In section 3 we go through the proposed approach. In section 4 we demonstrate the effectiveness of the proposed method on the evaluation datasets. Finally, section 5 addresses the challenges of the current approach, discusses future directions, and concludes the paper.

## 2 BACKGROUND & RELATED WORK

Koopman operator theory (Koopman, 1931) offers a linear, though infinite-dimensional, framework for studying nonlinear dynamical systems. By acting on observable functions, the Koopman operator enables a linear representation of nonlinear dynamics in a lifted space. Traditional methods, such as dynamic mode decomposition (DMD) (Schmid, 2010; Schmid et al., 2011) and extended dynamic mode decomposition (EDMD) (Williams et al., 2015; Li et al., 2017), utilize a predefined library of functions to map the system's state to the observable space, where the dynamics can be approximated by the Koopman operator. However, a key challenge remains in selecting suitable observables. To overcome this, recent advancements incorporate deep learning techniques to automatically learn observable functions (Lusch et al., 2018; Otto & Rowley, 2019; Yeung et al., 2019), with approaches like deep-Koopman and deep-DMD dramatically enhancing the efficiency and scope of Koopman-based analysis (Yeung et al., 2019; Takeishi et al., 2017).

Recent studies such as Proctor et al. (2016) have expanded the Koopman operator's application into control systems and robotics , where it has shown promise in mapping nonlinear systems into linear representations, thus enabling real-time control strategies such as LQR and MPC (Mamakoukas et al., 2021; Korda & Mezić, 2018; Abraham et al., 2017). Learning-based methods further enhance the operator's performance by optimizing the embedding functions for complex nonlinear systems. This has led to improvements in control accuracy for systems like soft robots (Bruder et al., 2020; Shi & Meng, 2022) and vehicular applications (Cibulka et al., 2020; Wang et al., 2021), underscoring the Koopman operator's potential in a variety of domains. These developments continue to push the boundaries of how nonlinear systems can be modeled and controlled in real-world applications.

**Uncertainty quantification (UQ)** in deep learning is essential for assessing the reliability of predictions, especially in critical domains such as autonomous systems and scientific applications. UQ methods allow models to provide not only predictions but also estimates of how confident they are in these predictions. Techniques such as Bayesian neural networks, deep ensembles (Lakshminarayanan et al., 2017), Monte Carlo dropouts (Gal & Ghahramani, 2016), and evidential deep learning (Sensoy et al., 2018; Amini et al., 2020) are commonly used to estimate uncertainties, distinguishing between aleatoric uncertainty (stemming from noise in the data) and epistemic uncertainty (arising from model limitations) (Abdar et al., 2021). These methods help improve the robustness of deep learning models, making them more reliable for real-world decision-making tasks such as planning for autonomous vehicles.

In Koopman-based frameworks, uncertainty quantification is becoming increasingly important for modeling and controlling nonlinear dynamical systems. By incorporating uncertainty into the prediction of Koopman eigenfunctions or observables, as demonstrated by Morton et al. (2019), We can more effectively assess the reliability of system behavior predictions, especially in situations where the training data is noisy or incomplete. In the Koopman framework, methods such as Bayesian neural networks (Pan & Duraisamy, 2020) and deep ensembles (Frion et al., 2024) have been utilized to address uncertainty; however, these approaches are not suitable for real-time applications. Alternative strategies improve robustness in control and prediction tasks by introducing a probability distribution over the embedding space (observables) for the initial state and tracking its propagation over time (Han et al., 2021; Meyers et al., 2019). While effective, these methods fail to account for uncertainties in the underlying system dynamics. In contrast, our approach directly addresses this limitation by placing a probability distribution over the Koopman operator itself, offering a rigorous and comprehensive framework for quantifying both aleatoric and epistemic uncertainties. This integration of uncertainty quantification with Koopman operators enables a more robust modeling paradigm, paving the way for advanced data-driven control of nonlinear systems while explicitly accounting for inherent uncertainties.

**Distribution Shifts** refers to the condition where the statistical characteristics of the data vary specially at test time. These shifts can stem from evolving underlying processes, such as gradual temporal changes or sudden external events, leading to a mismatch between the data distribution a model was trained on and the new, unseen conditions it encounters. This issue is especially prevalent in dynamic systems, such as vehicle navigation in varying road conditions (e.g., icy, wet, or dry surfaces), where fluctuating environmental factors can significantly degrade model performance. Traditional machine learning models, which often assume stationary data distributions, typically struggle to generalize effectively across different time periods. To mitigate these challenges, some approaches, such as Passalis et al. (2019), adaptively stationarize the inputs. Others, for instance Arik et al. (2022), use test-time adaptation to tackle the distribution shifts problems. In the context of Koopman framework, other approaches, such as the Koopman Neural Forecaster (KNF) (Wang et al., 2022) that was proposed for time series forecasting, utilize a combination of global and local operators to capture both stable and evolving dynamics, allowing for continuous adaptation and enhanced robustness in the face of the non-stationary nature of real-world time series data. However, their approach is confined to time series forecasting and does not account for control inputs. In contrast, this work introduces a framework that systematically addresses uncertainty quantification and manages distribution shifts for dynamical systems with control inputs. Next, we present the methodology for the proposed approach in detail.

## 3 METHODOLOGY

In the following section, we begin by outlining the modeling problem. Then, we present the proposed architecture, followed by the Bayesian approach for learning the Koopman operator and quantifying uncertainty. Afterward, we discuss the online adaptation of the learned operator. Finally, we go through how the proposed method can be extended to allow real-time motion planning.

### 3.1 KOOPMAN THEORY FOR MODELING NONLINEAR DYNAMICAL SYSTEMS

An autonomous vehicle can be represented as a nonlinear dynamical system. Consider a physical system whose state at time $t$ is denoted by $x_t \in \mathbb{R}^n$. The evolution of the system can be modeled as a discrete-time dynamical system:

$$x_{t+1} = f(x_t, u_t) \tag{1}$$

where $u_t \in \mathbb{R}^m$ represents the external inputs or control actions applied to the system at time $t$, and $f$ is a nonlinear function that describes the system's dynamics. For a future time horizon $h$, the objective is to predict the future states of the system $x_{t+1}, x_{t+2}, \ldots, x_{t+h}$, given the sequence of current and past states $x_t, x_{t-1}, \ldots, x_{t-q}$ and past and future inputs $u_{t+h}, \ldots, u_t, \ldots, u_{t-q}$ for a history window of size $q$.

The Koopman operator, denoted as $\bar{\mathcal{K}} : \bar{\mathcal{F}} \to \bar{\mathcal{F}}$, is an infinite-dimensional linear operator that describes the evolution of a nonlinear dynamical system. Here, $\bar{\mathcal{F}}$ refers to the set of all *measurement functions* or *observables*, which form an infinite-dimensional Hilbert space. More specifically, given an observable function $\psi$, the evolution of the system using the Koopman operator can expressed as:

$$\bar{\mathcal{K}}\psi(x_t, u_t) = \psi(f(x_t, u_t), u_{t+1}) = \psi(x_{t+1}, u_{t+1}) \tag{2}$$

The primary challenge in utilizing the Koopman operator lies in identifying appropriate observable functions that encapsulate the key dynamics of the system. To address this, we seek to identify a subspace $\mathcal{F} \subset \bar{\mathcal{F}}$ that approximately preserves invariance under the Koopman operator. This subspace is spanned by a set of linearly independent basis functions, $g : \mathbb{R}^n \times \mathbb{R}^m \to \mathbb{R}^d$, which provide a finite-dimensional approximation of the Koopman operator, denoted as $\mathcal{K}$. This finite-dimensional operator advances the observable function $g$ over time.

Applying these observables to the system's state and control inputs at time $t$ produces an embedding vector $\tilde{z}_t \in \mathbb{R}^d$, which maps the original state $x_t$ and control input $u_t$ into a higher-dimensional space, where $d \gg n + m$. Specifically, we have $\tilde{z}_t = g(x_t, u_t)$. This embedding vector can be decomposed into two parts: $\tilde{x}_t \in \mathbb{R}^\eta$, representing the state embedding, and $\tilde{u}_t \in \mathbb{R}^{d-\eta}$, representing the control embedding. The system's dynamics can then be approximated using the linear operator $\mathcal{K} \in \mathbb{R}^{\eta \times d}$, which provides a finite-dimensional approximation of the Koopman operator. Specifically, given an embedding $\tilde{z}_t$, the evolution of the system under the Koopman operator can be expressed as:

$$\tilde{x}_{t+1} = \mathcal{K}\tilde{z}_t \tag{3}$$

This linear representation in the embedding space allows us to forecast future system states. By learning the appropriate embedding function $g$, we can effectively capture the nonlinear dynamics of the system in a linear framework, facilitating the prediction of future states by evolving the embedding forward in time.

Previous methods for projecting the state-action space $[x_t, u_t]^T$ into the Koopman embedding space $\tilde{z}_t$ typically process each time step independently. In contrast, we adopt a more general approach by modeling the embeddings as a sequence that depends on consecutive state-action pairs. Specifically, we aim to obtain a sequence of embedding vectors as follows:

$$[\tilde{z}_{t-q} \quad \ldots \quad \tilde{z}_{t-1} \quad \tilde{z}_t] = G_\theta(x_{t-q}, u_{t-q}, \ldots, x_{t-1}, u_{t-1}, x_t, u_t) \tag{4}$$

where $G_\theta$, the embedding function, is implemented as a neural network parameterized by $\theta$. Unlike previous works that perform embeddings for individual states, we construct embeddings for an entire trajectory, as outlined in Eq. 4. Specifically, the embedding $\tilde{z}_t$ is not solely derived from the current state-action pair $[x_t, u_t]^T$, but also incorporates delayed state-action pairs. This formulation enables the embedding vectors to capture dynamic relationships across multiple time steps, providing contextual information for each embedding. A key motivation for this approach is rooted in Takens's theorem (Takens, 2006), which suggests that the use of delayed coordinates can capture the underlying system dynamics more accurately. Empirically, we demonstrate that this method yields richer and higher-quality embeddings. Additionally, this formulation facilitates an online model adaptation using previously computed embeddings, a capability that will be further elaborated upon in subsequent sections. The trajectory of consecutive states and actions is encoded using a transformer encoder, which we refer to as the *trajectory encoder* (see Figure 1 for details).

To enable multi-step predictions, embeddings of future control inputs $u_{t+1}, u_{t+2}, \ldots, u_{t+h}$ are obtained using a dedicated encoder. These future input embeddings are conditioned on the current trajectory, providing a consistent dynamical context to improve embedding quality. This is accomplished through a transformer decoder, where the cross-attention mechanism allows the future action

Figure 1: The proposed transformer architecture follows this sequence: (1) Encode past state-action pairs with a trajectory encoder to produce Koopman embeddings. (2) These embeddings, except for the current time step, are utilized to update the estimated Koopman operator and observation matrix distributions. (3) Predict the next future state embedding by combining the current time-step embedding with these updated distributions. This future state embedding is then concatenated with action embeddings and then propagated using the Koopman operator iteratively. The probabilistic observation matrix converts the state embeddings into the predicted future states.

embeddings to incorporate information from prior state-action pairs, enhancing the representation of future actions. We refer to this module as the *action encoder* (illustrated in Figure 1). Finally, for mapping from the Koopman latent space to the state space, we use a linear decoder to retrieve the reconstructed state $\hat{x}_t$ from the state embedding $\tilde{x}_t$ as follows:

$$\hat{x}_t = \mathcal{C}\tilde{x}_t \tag{5}$$

The loss function at each time-step $t$ consists of two key components: the alignment loss, which ensures that the embeddings are properly aligned through linear system dynamics, and the prediction loss, which focuses on accurately reconstructing the state. These losses are formally expressed as follows:

$$\mathcal{L}_{\text{Align}} = \sum_{i=-q+1}^{0} \|\tilde{x}_{t+i} - \mathcal{K}\tilde{z}_{t+i-1}\|_2, \quad \mathcal{L}_{\text{Pred}} = \sum_{i=-q}^{h} \|x_{t+i} - \mathcal{C}\tilde{x}_{t+i}\|_2 \tag{6}$$

## 3.2 BAYESIAN LEARNING OF ADAPTIVE STOCHASTIC KOOPMAN OPERATOR

To address system noise and uncertainty, we incorporate Bayesian modeling into the proposed architecture. Our objective is to apply Bayesian techniques to both the Koopman operator and the observation matrix in the embedding space. Our method follows a curriculum training approach: we begin by training the trajectory and action encoders to produce the corresponding embeddings, which are then used, along with ground truth data, within a Bayesian learning framework. Specifically, the learned embeddings over all timesteps and trajectories create a dataset of temporal state transitions $\tilde{z}_t \to \tilde{x}_{t+1}$ and state reconstruction pairs $\tilde{x}_t \to x_t$. It is important to note that the ground truth for state reconstruction corresponds to the actual system state. Using the same notation for the embeddings, the noise-adaptive system can then be described as follows:

$$\tilde{x}_{t+1}^i = \mathcal{K}\tilde{z}_t^i + \epsilon^i \tag{7}$$

$$x_t^i = \mathcal{C}\tilde{x}_t^i + \upsilon^i \tag{8}$$

where $i$ indexes each datapoint or transition in the dataset, $\mathcal{K}, \mathcal{C}$ are the stochastic Koopman and observation matrices, and the noise terms, $\epsilon^i \sim \mathcal{N}(0, \Sigma)$ and $\upsilon^i \sim \mathcal{N}(0, \Phi)$, represent process and observation noise, assumed to follow Gaussian distributions with covariances $\Sigma$ and $\Phi$ respectively.

Our objective is to learn two Bayesian regression models for both the stochastic Koopman operator, and the observation matrix. For brevity, we will focus solely on modeling the stochastic Koopman operator $\mathcal{K}$, as the approach for the reconstruction (observation) matrix is analogous. Under the assumption that the noise vectors are i.i.d sampled from a multivariate Gaussian distribution, the likelihood of transitioning from $\tilde{z}_t^i$ to $\tilde{x}_{t+1}^i$, given $\mathcal{K}$ and the covariance $\Sigma$, is:

$$p(\tilde{X}|\mathcal{K}, \tilde{Z}, \Sigma) = \frac{1}{(2\pi)^{\eta d}|\Sigma|^{\frac{\eta}{2}}} \exp\left(-\frac{1}{2}\sum_{i=1}^{N}\left(\left(\tilde{x}_{t+1}^i - \mathcal{K}\tilde{z}_t^i\right)^\top \Sigma^{-1}\left(\tilde{x}_{t+1}^i - \mathcal{K}\tilde{z}_t^i\right)\right)\right) \quad (9)$$

where $\tilde{X} \in \mathbb{R}^{\eta \times N}$, $\tilde{Z} \in \mathbb{R}^{d \times N}$, $\Sigma \in \mathbb{R}^{\eta \times \eta}$, where $N$ is the number of data points. The conjugate prior for this likelihood is the matrix normal inverse Wishart distribution $\mathcal{MNIW}$:

$$\mathcal{K}, \Sigma \sim \mathcal{MNIW}(\breve{M}, \breve{V}, \breve{\nu}, \breve{\Psi}), \quad (10)$$

where $\mathcal{K}|\Sigma \sim \mathcal{MN}(\breve{M}, \Sigma, \breve{V})$, meaning that given the covariance $\Sigma$, the Koopman operator $\mathcal{K}$ follows a matrix normal distribution with mean $\breve{M}$, and column covariance $\breve{V}$. The covariance $\Sigma \sim \mathcal{IW}(\breve{\nu}, \breve{\Psi})$ follows an inverse Wishart distribution, with parameters $\breve{\nu}$ and $\breve{\Psi}$, where the former is the degrees of freedom and the latter is the scale matrix.

**Lemma 3.1.** *Given the likelihood (Eq.9), and the ($\mathcal{MNIW}$) prior (Eq.(10))) , the posterior distribution of the stochastic koopman operator $\mathcal{K}$ follows $\mathcal{MNIW}$ distribution and can be given by:*

$$\mathcal{K}, \Sigma \sim \mathcal{MNIW}(\hat{M}, \hat{V}, \hat{\nu}, \hat{\Psi}), \quad (11)$$

*with the posterior parameters given by:*

$$\hat{M} = S_{xz}S_{zz}^{-1}, \qquad \hat{V} = S_{zz}, \qquad \hat{\nu} = N + \breve{\nu}, \qquad \hat{\Psi} = \breve{\Psi} + S_{x|z} \quad (12)$$

*and*

$$S_{xz} = \tilde{X}\tilde{Z}^\top + \breve{M}\breve{V}, \quad S_{zz} = \tilde{Z}\tilde{Z}^\top + \breve{V}, \quad S_{xx} = \tilde{X}\tilde{X}^\top + \breve{M}\breve{V}\breve{M}^\top,$$
$$S_{x|z} = S_{xx} - S_{xz}S_{zz}^{-1}S_{xz}^\top \quad (13)$$

*Proof.* The proof follows directly from Murphy (2023). $\qquad \square$

**Lemma 3.2.** *Given the posterior distribution (Eq. 11) of $\mathcal{K}$ from Lemma 3.1, the posterior predictive distribution for the state transition at time $t+1$, conditioned on the current state and control inputs at time $t$, under the assumption that the number of datapoints $N$ is large, follows a multivariate Gaussian distribution:*

$$\tilde{x}_{t+1}|\tilde{z}_t, \tilde{X}, \tilde{Z} \sim \mathcal{N}\left(\mathcal{K}\tilde{z}_t, \hat{\Psi}\left(1 + \tilde{z}_t^\top \hat{V}\tilde{z}_t\right)\right) \quad (14)$$

*and the multi-step predictions for future states can be recursively computed as:*

$$\tilde{x}_{t+k}|\tilde{z}_t, \tilde{X}, \tilde{Z} \sim \mathcal{N}\left(\mathcal{K}^h\tilde{z}_t, \sum_{i=0}^{h-1}\mathcal{K}^i\left(\hat{\Psi}\left(1 + \tilde{z}_t^\top \hat{V}\tilde{z}_t\right)\left(\mathcal{K}^i\right)^\top\right)\right) \quad (15)$$

*Proof.* The proof can be found in Appendix. $\qquad \square$

**Adapting the Operator Online** A significant challenge in our application is handling *distribution shifts*—situations where the data encountered during deployment deviates from the data used in initial offline training. To address this, we leverage a "change variable," $s$, which detects distribution changes based on recent observations. Rather than using a single observation, a history window of size $q$ is examined (as shown by the trajectory encoder in Fig. 1), enabling the model to distinguish between shifts that require adaptation and transient noise, leading to a more robust update strategy.

In Bayesian modeling, the posterior distribution is computed by combining prior beliefs with new data. For online model updates, we treat the old, offline-trained model as the prior. As new data

becomes available, the posterior distribution is updated according to (Eq. 11), which defines how we combine this prior with the likelihood of the new observations to form the updated posterior. However, because of the extensive offline dataset, directly using the offline model as the prior can result in an overconfident prior. Consequently, the posterior distribution becomes dominated by the prior, making it challenging for new data to have any meaningful impact which results in slow adaptation. To mitigate this problem, we apply a *tempering operation* (Li et al., 2021) which systematically increases the prior variance, effectively broadening the prior distribution.

Tempering allows the model to "forget" outdated information while maintaining the flexibility to learn from new data. The tempering process is formalized by scaling the prior covariance by a factor $\beta^{-1}$, where $0 < \beta < 1$. Specifically, the tempered prior depends on whether a change is detected. If no change is detected ($s = 0$), the prior remains as it is. However, when a change is detected ($s_t = 1$), the prior is broadened using the temperature parameter $\beta$ as follows:

$$p(\mathcal{K} \mid s = 1, \Sigma) = \mathcal{MN}(\hat{M}, \beta^{-1}\Sigma, \beta^{-1}\hat{V}), \tag{16}$$

By increasing the prior variance, the model reduces its confidence in prior data, allowing it to more effectively learn from new, potentially shifted distributions.

For detecting changes, the update procedure is guided by the posterior of the change variable $s$, which is modeled as a Bernoulli distribution. Following Li et al. (2021), the probability of a change is determined through a likelihood ratio test, which compares the likelihood of the current embedding $z_t$ under the assumption of a change ($s = 1$) against the likelihood assuming no change ($s = 0$). The decision rule is formalized by the equation:

$$p(s = 1|\tilde{z}_{t-1:t-q}) = \sigma\left(\log\frac{p(\tilde{z}_t|s = 1)}{p(\tilde{z}_t|s = 0)} + \vartheta\right) \tag{17}$$

where $\sigma$ is the sigmoid function, and $\vartheta$ is a hyperparameter favoring either change or no change. This change detection mechanism allows for precise and timely model updates, ensuring the model remains responsive to dynamic environments.

### 3.3 INTEGRATION INTO MOTION PLANNING

To effectively integrate the proposed method into a sampling-based motion planner, it is essential to generate a large number of trajectories by sampling from the action space. This process enables the creation of dynamically feasible trajectories in real time. However, in our current framework, sampling actions requires computing their corresponding Koopman embeddings through the *action encoder*. This step is computationally intensive due to the overhead of encoding each sampled action.

To address this limitation, we transform the action encoder into a *variational action encoder*. The primary goal of this modification is to learn a normalized Gaussian distribution over the embedding space, enabling direct sampling from this distribution without the need to pass actions through the encoder during the planning phase. The variational encoder modifies the deterministic mapping of the traditional encoder into a probabilistic one by mapping each input action to a distribution over embeddings, parameterized by a zero mean vector and a unit standard deviation vector. This allows us to sample embeddings directly making the trajectory generation process and improving computational efficiency.

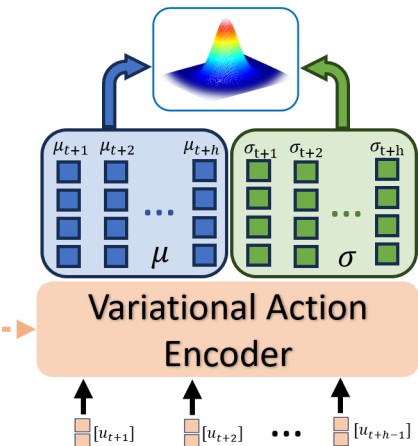

Figure 2: We extend the previous *action encoder* into a *variational encoder* to learn a Gaussian distribution over the embedded action space. This enables us to sample directly from the latent space at runtime, bypassing the encoding step entirely.

To ensure that the learned embeddings approximate a Gaussian distribution with zero mean and unit standard deviation, we incorporate a Kullback-Leibler (KL) divergence term into the loss function

(Eq. 6) during training. By minimizing this divergence, we encourage the *action encoder* to produce embeddings that closely match the standard normal distribution. This alignment allows for efficient and direct sampling of embeddings during the planning phase, as we can sample directly from the standard normal distribution without relying on the encoder. Consequently, this modification enhances the real-time capabilities of the motion planner through more efficient trajectory generation.

# 4 EXPERIMENTAL RESULTS

In this section, we assess the performance of the proposed method across various highly nonlinear environments, characterized by different dimensionalities and noise levels, including a truck dynamics dataset under diverse weather conditions. We start by detailing the environments and baseline models, followed by an evaluation of the prediction accuracy. Next, we analyze the uncertainty quantification, and finally, provide an example of the method's application in planning tasks.

**Evaluation Datasets.** The primary dataset used for evaluation is a truck and trailer dataset, specifically aimed at learning an accurate dynamic model for a 37.5-ton, 17-meter-long Scania truck and trailer (see Fig. 3). Data collection was carried out both during autonomous driving and with assistance from a professional safety driver. State feedback signals were gathered from inertial and navigation sensors during tests conducted on various surfaces, including dry asphalt, wet roads, snow, and ice under winter conditions. To reduce high-frequency noise, the recorded state trajectories were processed using a 4th-order Butterworth lowpass filter with a 5 Hz cutoff frequency. All input features were scaled between [-1, 1] to ensure the neural network assigns equal importance to each data component. The state of the system is a seven dimensional vector containing the longitudinal and lateral velocities, longitudinal and lateral accelerations, yaw angle of the tractor, the trailer angle with the tractor, and finally, the slip angle of the front wheel. As for the inputs, they are the brake, thrust as well as the steering of the vehicle. Several environmental inputs are present such as road grade, estimated road type, and other vehicle related characteristics. The processed training dataset encompasses 10 hours of vehicle trajectory data, derived from real-world driving tests conducted between March 2023 and March 2024. These tests were designed to include a wide range of challenging scenarios, effectively capturing the intricate dynamics of trucks and trailers.

To ensure a robust and thorough evaluation of the model's performance, the test dataset was exclusively collected during the dedicated winter testing phase at Scania's proving ground in northern Sweden in the winter of 2024. This dataset emphasizes the unique challenges posed by harsh winter conditions, providing a critical evaluation of the model's ability to navigate real-world complexities in vehicle dynamics. It also serves as a highly realistic dataset that introduces distribution shifts not present in the training data, thereby enabling a more comprehensive assessment of the model's adaptability and robustness under previously unseen conditions. It includes diverse scenarios like forward driving, sharp turns, U-turns, roundabouts, steep ascents/descents, and mu-split conditions, all complicated by snow and ice. By focusing on winter-specific challenges, this dataset rigorously tests the model's robustness and adaptability under conditions impacting traction, stability, and control, providing valuable insights into performance in extreme scenarios. More details are in Appendix B.

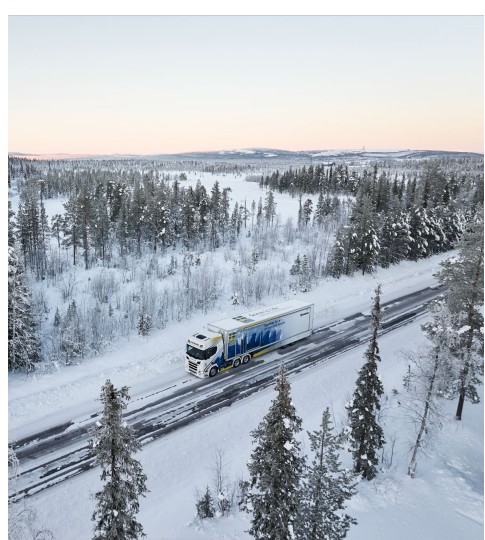

Figure 3: Truck and trailer heavy duty vehicle in winter conditions.

To assess scalability, we evaluated our method on five additional Gym environments simulated using the MuJoCo physics engine: Hopper, HalfCheetah, Ant, InvertedPendulum, and Walker. The data for these environments were collected using a TD3 expert agent trained for 1 million time steps before being utilized for data collection. Each environment was tested under three conditions: with

process noise, with observation noise, and without any noise. Further details about these environments and their configurations are provided in the appendix.

**Baselines.** We evaluate our method by comparing it against six baselines for both prediction accuracy and uncertainty estimation: (1) **Deep Stochastic Koopman Operator (Desko)** (Han et al., 2021), which uses an encoder to model a distribution over observable functions of the states and incorporates linear control. (2) **Deep Koopman Operator (DKO)** (Shi & Meng, 2022), which employs a neural network to jointly learn observable functions and the Koopman operator, embedding a single state-action pair rather than entire trajectories. (3) **EMLP**, an ensemble of 10 multi-layer perceptrons providing uncertainty estimation through prediction variance across models. (4) **Neural ODEs (NODE)**, (Chen et al., 2018) which use continuous-time dynamics modeled by neural networks to capture smooth trajectory behavior. (5) **MC Dropout**, where uncertainty is estimated by applying dropout at inference time across multiple forward passes. (6) **Bayesian Neural Networks (BNNs)**, which estimate prediction uncertainty by learning posterior distributions over model weights. Additionally, we evaluate our method in two configurations: one with a traditional action encoder and another with a variational action encoder with a small embedding dimension, to assess prediction performance in sampling-based planners.

**Training Setup.** To ensure a fair comparison, all models were configured with approximately 45,000 parameters, except for the ensemble of multi-layer perceptrons. Training was conducted over 300 epochs across various datasets, as this was sufficient for the baselines to converge. Since our approach (Blak) leverages a history window, it was provided with an additional 20 time-steps of history. To ensure robustness, each method was run 10 times using different random seeds across all environments. The prediction horizon for all methods was set to 200 time-steps, corresponding to a 10-second planning horizon at 20 Hz. This number reflects the frequency at which our planner operates. Additionally, to evaluate generalization beyond the training sequence length, methods were also tested on a prediction horizon of 300 time-steps to confirm that our method accurately learns the underlying dynamics. The objective for all methods was to minimize the same prediction error over these horizons. Training utilized a batch size of 1,024 and an initial learning rate of 0.003, which decayed by one-third after each third of the training epochs. Additional Information can be found in Appendix F.

**Prediction Evaluation.** For the truck dynamics dataset (additional simulated environments detailed in Appendix D), Table 1 reports the mean squared error for multi-step predictions on the validation set across all baselines. The proposed method demonstrates superior prediction accuracy compared to other baselines when utilizing a standard encoder. With a variational action encoder, the performance of our approach is comparable to that of DKO, while maintaining the capability to run in real-time—a significant advantage over DKO. Traditional baselines such as Bayesian NN and MC Dropout show higher loss, indicating difficulty in effectively modeling the underlying dynamics. Architectures like DKO and NODE exhibit moderate errors, but their larger standard deviations reveal inconsistencies in prediction quality. These results highlight the strength of the proposed architecture in delivering both accurate and stable predictions for complex systems. Notably, the variational action encoder can be scaled with additional parameters without compromising real-time performance, however, a similar number of parameters to other baselines was chosen for comparison.

| Baseline | Final Loss ($\pm$ Std) |
|---|---|
| MCDropout: | $0.4373 \pm 0.0867$ |
| Blak: | $\mathbf{0.1016 \pm 0.0519}$ |
| BayesianNN: | $0.5966 \pm 0.0354$ |
| Dko: | $0.1664 \pm 0.0932$ |
| NODE: | $0.2319 \pm 0.2020$ |
| Desko: | $0.4658 \pm 0.0597$ |
| EMLP: | $0.3064 \pm 0.0301$ |
| BlakVar: | $0.2002 \pm 0.0776$ |

Table 1: Final Loss (Mean $\pm$ Std) for Each Baseline.

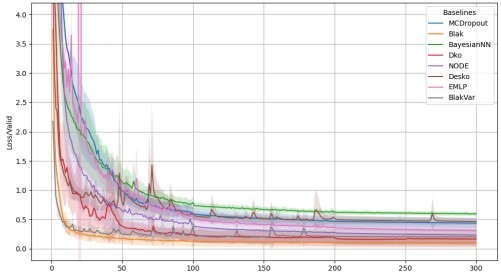

Figure 4: Validation Loss Across Baselines.

**Uncertainty Quantification.** The accuracy of our uncertainty estimation is evaluated by examining the correlation between the predicted uncertainty and the error rates. A strong correlation

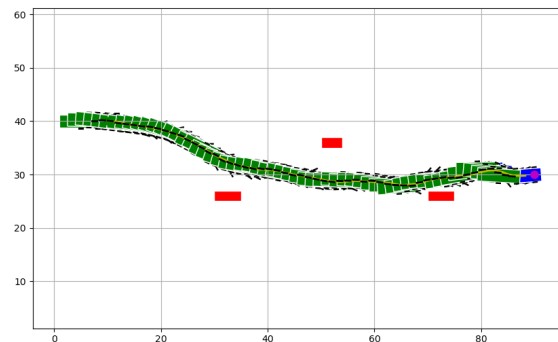

Figure 5: An example of a planned path using our dynamic model within an RRT framework. Actions are sampled from the embedding space, and future states are predicted by the model to evaluate trajectory costs, enabling dynamically accurate path planning.

suggests that the model expresses higher uncertainty when its predictions are less accurate, which is a desirable characteristic. To test this, we conducted a correlation analysis between the predicted uncertainty from each model and the error rates on 100,000 randomly selected points across various trajectories and time-steps. As shown in Table 2, our model, along with EMLP, exhibits strong correlation values, while the Desko method underperforms compared to both.

Table 2: Average correlation values of uncertainty with MSEs for each dataset. We see that both EMLP and BLAK exhibit a strong correlation between prediction error and predicted uncertainty.

| Dataset | Desko | EMLP | BNN | MC Dropout | Blak (ours) |
|---|---|---|---|---|---|
| Truck | .43 | .68 | 0.56 | 0.48 | **.71** |
| Walker | .39 | .59 | 0.53 | 0.45 | **.67** |
| Ant | .48 | .64 | 0.52 | 0.49 | **.73** |
| Half-Cheetah | .46 | **.69** | 0.63 | 0.43 | **.69** |
| Hopper | .47 | .53 | 0.29 | 0.27 | **.64** |

**Application to Motion Planning.** Finally, we demonstrate the applicability of our models in dynamically-aware planning. To integrate the proposed architecture into sampling-based planning, we employ the learned vehicle dynamics model within a Rapidly-exploring Random Tree (RRT) algorithm, with a key modification: we sample from the action space instead of the state space. By applying the learned dynamic model to these sampled actions, we compute new states. We then apply cost functions to the resulting transitions, leading to dynamically accurate planned paths. Sampling directly from the action embedding space allows us to efficiently generate new nodes in the RRT algorithm by simply multiplying the Koopman matrix and the observation matrix with the corresponding embeddings. To further enhance planning efficiency, we compress the action embedding space to just two dimensions, making sampling much more efficient. Figure 5 presents a visualization of the results, demonstrating the effectiveness of the proposed method in sampling-based planning. Additional information cab be found in the Appendix.

## 5 CONCLUSION & FUTURE WORK

In this work, we propose a Bayesian learning approach for an adaptive Koopman operator to model vehicle dynamics while incorporating uncertainty estimation. Additionally, we showcase the application of this method in sampling-based motion planning. Our approach offers accurate predictions while maintaining real-time performance, and it scales efficiently, allowing for larger network sizes with minimal impact on inference time. When applied to motion planning, it not only ensures real-time execution but also generates more accurate, dynamic-aware paths that respect the vehicle's physical constraints. For future work, we aim to further explore this approach within the planning domain and investigate ways to reduce the gap between variational and nominal inference. Additionally, the model's uncertainty estimation capabilities make it a promising candidate for model-based reinforcement learning, where it could be leveraged to guide exploration and learn world models for adaptive decision-making in stochastic environments.

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

## A  POSTERIOR PREDICTIVE DISTRIBUTION

Given an embedding $\tilde{z}_t$ in time $t$, the goal is to predict the probability over the state embedding for the next time-step $p(\tilde{x}_{t+1}|\tilde{z}_t, \mathcal{D})$, where $\mathcal{D}$ denotes the offline dataset.

$$p(\tilde{x}_{t+1}|\tilde{z}_t, \mathcal{D}) = \iint p(\tilde{x}_{t+1}|\mathcal{K}, \Sigma, \tilde{z}_t, \mathcal{D})p(\mathcal{K}|\Sigma)p(\Sigma) \quad d\mathcal{K}d\Sigma \tag{18}$$

First, we look at $\int p(\tilde{x}_{t+1}|\mathcal{K}, \Sigma, \tilde{z}_t, \mathcal{D})p(\mathcal{K}|\Sigma)d\mathcal{K}$. We know that:

$$p(\mathcal{K}|\Sigma) = \mathcal{MN}(\hat{M}, \Sigma, \hat{V})$$

$$= \frac{1}{(2\pi)^{\frac{\eta k}{2}}|\Sigma|^{\frac{k}{2}}|\hat{V}|^{\frac{\eta}{2}}} exp\left[\frac{-1}{2}tr\left(\left(\mathcal{K} - \hat{M}\right)^{\top}\Sigma^{-1}\left(\mathcal{K} - \hat{M}\right)\hat{V}^{-1}\right)\right] \tag{19}$$

and,

$$p(\tilde{x}_{t+1}|\mathcal{K}, \Sigma, \tilde{z}_t, \mathcal{D}) = \frac{1}{(2\pi)^{\frac{\eta}{2}}|\Sigma|^{\frac{1}{2}}} exp\left[\frac{-1}{2}(\tilde{x}_{t+1} - \mathcal{K}\tilde{z}_t)^{\top}\Sigma^{-1}(\tilde{x}_{t+1} - \mathcal{K}\tilde{z}_t)\right] \tag{20}$$

Then:

$$\int p(\tilde{x}_{t+1}|\mathcal{K}, \Sigma, \tilde{z}_t, \mathcal{D})p(\mathcal{K}|\Sigma)d\mathcal{K}$$

$$= \int \frac{1}{(2\pi)^{\frac{\eta+\eta k}{2}}|\Sigma|^{\frac{1+k}{2}}|\hat{V}|^{\frac{\eta}{2}}} exp\left[\frac{-1}{2}tr\left(\Sigma^{-1}\left((\tilde{x}_{t+1} - \mathcal{K}\tilde{z}_t)(\tilde{x}_{t+1} - \mathcal{K}\tilde{z}_t)^{\top} + \left(\mathcal{K} - \hat{M}\right)\hat{V}^{-1}\left(\mathcal{K} - \hat{M}\right)^{\top}\right)\right)\right]$$

$$= \int \frac{1}{(2\pi)^{\frac{\eta+\eta k}{2}}|\Sigma|^{\frac{1+k}{2}}|\hat{V}|^{\frac{\eta}{2}}} exp\left[\frac{-1}{2}tr\left(\Sigma^{-1}\left(\mathcal{K}\left(\tilde{z}_t\tilde{z}_t^{\top} + \hat{V}^{-1}\right)\mathcal{K}^{\top} - 2\left(\tilde{x}_{t+1}\tilde{z}_t^{\top} + \hat{M}\hat{V}^{-1}\right)\mathcal{K}^{\top}\right.\right.\right.$$

$$\left.\left.\left. + \hat{M}\hat{V}^{-1}\hat{M}^{\top} + \tilde{x}_{t+1}(\tilde{x}_{t+1})^{\top}\right)\right)\right]$$

$$= \int \frac{1}{(2\pi)^{\frac{\eta+\eta k}{2}}|\Sigma|^{\frac{1+k}{2}}|\hat{V}|^{\frac{\eta}{2}}} exp\left[\frac{-1}{2}tr\left(\Sigma^{-1}\left(\mathcal{K}S_{aa}\mathcal{K}^{\top} - 2S_{ab}\mathcal{K}^{\top} + S_{bb}\right)\right)\right]$$

$$= \int \frac{1}{(2\pi)^{\frac{\eta+\eta k}{2}}|\Sigma|^{\frac{1+k}{2}}|\hat{V}|^{\frac{\eta}{2}}} exp\left[\frac{-1}{2}tr\left(\Sigma^{-1}\left(\left(\mathcal{K} - S_{ab}S_{aa}^{-1}\right)S_{aa}\left(\mathcal{K} - S_{ab}S_{aa}^{-1}\right)^{\top} + S_{a|b}\right)\right)\right]$$

$$= \frac{(2\pi)^{\frac{k\eta}{2}}|\Sigma|^{\frac{k}{2}}|S_{aa}|^{-\frac{\eta}{2}}}{(2\pi)^{\frac{\eta+\eta k}{2}}|\Sigma|^{\frac{1+k}{2}}|\hat{V}|^{\frac{\eta}{2}}} exp\left[\frac{-1}{2}tr\left(\Sigma^{-1}S_{a|b}\right)\right] = \frac{|S_{aa}|^{-\frac{\eta}{2}}}{(2\pi)^{\frac{\eta}{2}}|\Sigma|^{\frac{1}{2}}|\hat{V}|^{\frac{\eta}{2}}} exp\left[\frac{-1}{2}tr\left(\Sigma^{-1}S_{a|b}\right)\right] \tag{21}$$

where:

$$S_{aa} = \tilde{z}_t\tilde{z}_t^{\top} + \hat{V}^{-1}$$
$$S_{ab} = \tilde{x}_{t+1}\tilde{z}_t^{\top} + \hat{M}\hat{V}^{-1}$$
$$S_{bb} = \hat{M}\hat{V}^{-1}\hat{M}^{\top} + \tilde{x}_{t+1}(\tilde{x}_{t+1})^{\top}$$
$$S_{a|b} = S_{bb} - S_{ab}S_{aa}^{-1}S_{ab}^{\top}$$

Substituting Eq. 21 into Eq. 18 yields:

$$p(\tilde{x}_{t+1}|\tilde{z}_t, \mathcal{D}) = \int \frac{|S_{aa}|^{-\frac{\eta}{2}}|\hat{\Psi}|^{\frac{\hat{\nu}}{2}}}{(2\pi)^{\frac{\eta}{2}}|\Sigma|^{\frac{1}{2}}|\hat{V}|^{\frac{\eta}{2}}2^{\frac{\hat{\nu}\eta}{2}}\Gamma_\eta\left(\frac{\hat{\nu}}{2}\right)}|\Sigma|^{-\frac{\hat{\nu}+\eta+1}{2}}exp\left[\frac{-1}{2}tr\left(\Sigma^{-1}\left(S_{a|b}+\hat{\Psi}\right)\right)\right] \qquad d\Sigma$$

$$= \frac{|S_{aa}|^{-\frac{\eta}{2}}|\hat{\Psi}|^{\frac{\hat{\nu}}{2}}}{(2\pi)^{\frac{\eta}{2}}|\hat{V}|^{\frac{\eta}{2}}2^{\frac{\hat{\nu}\eta}{2}}\Gamma_\eta\left(\frac{\hat{\nu}}{2}\right)}\int|\Sigma|^{-\frac{\hat{\nu}+\eta+1+1}{2}}exp\left[\frac{-1}{2}tr\left(\Sigma^{-1}\left(S_{a|b}+\hat{\Psi}\right)\right)\right] \qquad d\Sigma$$

$$= \frac{|S_{aa}|^{-\frac{\eta}{2}}|\hat{\Psi}|^{\frac{\hat{\nu}}{2}}}{(2\pi)^{\frac{\eta}{2}}|\hat{V}|^{\frac{\eta}{2}}2^{\frac{\hat{\nu}\eta}{2}}\Gamma_\eta\left(\frac{\hat{\nu}}{2}\right)}\frac{2^{\frac{\hat{\nu}\eta}{2}}\Gamma_\eta\left(\frac{\hat{\nu}+\eta}{2}\right)}{|S_{a|b}+\hat{\Psi}|^{\frac{\hat{\nu}+\eta}{2}}}$$

$$= \frac{\Gamma_\eta\left(\frac{\hat{\nu}+\eta}{2}\right)|S_{aa}|^{-\frac{\eta}{2}}|\hat{\Psi}^{-\frac{\eta}{2}}|}{(2\pi)^{\frac{\eta}{2}}\Gamma_\eta\left(\frac{\hat{\nu}}{2}\right)|\hat{V}|^{\frac{\eta}{2}}}|I+\hat{\Psi}^{-1}S_{a|b}|^{-\frac{\hat{\nu}+\eta}{2}} \qquad (22)$$

Now, let's break $S_{a|b}$ first. We know that:

$$S_{a|b} = S_{bb} - S_{ab}S_{aa}^{-1}S_{ab}^\top$$

$$= \hat{M}\hat{V}^{-1}\hat{M}^\top + \tilde{x}_{t+1}(\tilde{x}_{t+1})^\top - \left(\tilde{x}_{t+1}\tilde{z}_t^\top + \hat{M}\hat{V}^{-1}\right)\underbrace{\left(\tilde{z}_t\tilde{z}_t^\top + \hat{V}^{-1}\right)^{-1}}_{C}\left(\tilde{x}_{t+1}\tilde{z}_t^\top + \hat{M}\hat{V}^{-1}\right)^\top$$

$$= \hat{M}\hat{V}^{-1}\hat{M}^\top + \tilde{x}_{t+1}(\tilde{x}_{t+1})^\top - \tilde{x}_{t+1}\tilde{z}_t^\top C\tilde{z}_t(\tilde{x}_{t+1})^\top + \tilde{x}_{t+1}\tilde{z}_t^\top C\hat{V}^{-1}\hat{M}^\top + \hat{M}\hat{V}^{-1}C\tilde{z}_t(\tilde{x}_{t+1})^\top + \hat{M}\hat{V}^{-1}C\hat{V}^{-1}\hat{M}^\top$$

$$= \tilde{x}_{t+1}\underbrace{\left(I - \tilde{z}_t^\top C\tilde{z}_t\right)}_{S_{ii}}(\tilde{x}_{t+1})^\top - 2\underbrace{\hat{M}\hat{V}^{-1}C\tilde{z}_t}_{S_{ij}}(\tilde{x}_{t+1})^\top + \underbrace{\hat{M}\hat{V}^{-1}\hat{M}^\top + \hat{M}\hat{V}^{-1}C\hat{V}^{-1}\hat{M}^\top}_{S_{jj}}$$

$$= \left(\tilde{x}_{t+1} - S_{ij}S_{ii}^{-1}\right)^\top S_{ii}\left(\tilde{x}_{t+1} - S_{ij}S_{ii}^{-1}\right) + \underbrace{S_{j|i}}_{=0} \qquad (23)$$

Now, we have reached a quadratic formula. We begin by using the Woodbury formula on $S_{ii}^{-1}$:

$$S_{ii}^{-1} = \left(I - \tilde{z}_t^\top C\tilde{z}_t\right)^{-1}$$

$$= I + \tilde{z}_t^\top\left(C^{-1} - \tilde{z}_t\tilde{z}_t^\top\right)^{-1}\tilde{z}_t$$

$$= I + \tilde{z}_t^\top\left(\tilde{z}_t\tilde{z}_t^\top + \hat{V}^{-1} - \tilde{z}_t\tilde{z}_t^\top\right)^{-1}\tilde{z}_t$$

$$= 1 + \tilde{z}_t^\top\hat{V}\tilde{z}_t \qquad (24)$$

Then,

$$S_{ij}S_{ii}^{-1} = \hat{M}\hat{V}^{-1}C\tilde{z}_t\left(1 + \tilde{z}_t^\top\hat{V}\tilde{z}_t\right)$$

$$= \hat{M}\hat{V}^{-1}\left(\hat{V} - \hat{V}\tilde{z}_t\left(1 + \tilde{z}_t^\top\hat{V}\tilde{z}_t\right)^{-1}\tilde{z}_t^\top\hat{V}\right)\tilde{z}_t\left(1 + \tilde{z}_t^\top\hat{V}\tilde{z}_t\right)$$

$$= \hat{M}\left(I - \tilde{z}_t\left(1 + \tilde{z}_t^\top\hat{V}\tilde{z}_t\right)^{-1}\tilde{z}_t^\top\hat{V}\right)\tilde{z}_t\left(1 + \tilde{z}_t^\top\hat{V}\tilde{z}_t\right)$$

$$= \hat{M}\left(\tilde{z}_t\left(1 + \tilde{z}_t^\top\hat{V}\tilde{z}_t\right) - \tilde{z}_t\left(1 + \tilde{z}_t^\top\hat{V}\tilde{z}_t\right)^{-1}\tilde{z}_t^\top\hat{V}\tilde{z}_t\left(1 + \tilde{z}_t^\top\hat{V}\tilde{z}_t\right)\right)$$

$$= \hat{M}\left(\tilde{z}_t\left(1 + \tilde{z}_t^\top\hat{V}\tilde{z}_t\right) - \tilde{z}_t\tilde{z}_t^\top\hat{V}\tilde{z}_t\left(1 + \tilde{z}_t^\top\hat{V}\tilde{z}_t\right)^{-1}\left(1 + \tilde{z}_t^\top\hat{V}\tilde{z}_t\right)\right)$$

$$= \hat{M}\tilde{z}_t \qquad (25)$$

Substituting 25 and 23 into 22 yields:

$$p(\tilde{x}_{t+1}|\tilde{z}_t, \mathcal{D}) = \frac{\Gamma_\eta\left(\frac{\hat{\nu}+\eta}{2}\right)|S_{aa}|^{-\frac{\eta}{2}}|\hat{\Psi}|^{-\frac{\eta}{2}}}{(2\pi)^{\frac{\eta}{2}}\Gamma_\eta\left(\frac{\hat{\nu}}{2}\right)|\hat{V}|^{\frac{\eta}{2}}}|I + \hat{\Psi}^{-1}\left(\tilde{x}_{t+1} - \hat{M}\tilde{z}_t\right)^\top \left(1 + \tilde{z}_t^\top \hat{V}\tilde{z}_t\right)^{-1}\left(\tilde{x}_{t+1} - \hat{M}\tilde{z}_t\right)|^{-\frac{\hat{\nu}+\eta}{2}}$$

Using the matrix determinant lemma, this equals to:

$$p(\tilde{x}_{t+1}|\tilde{z}_t, \mathcal{D}) = \frac{\Gamma_\eta\left(\frac{\hat{\nu}+\eta}{2}\right)}{(2\pi)^{\frac{\eta}{2}}\Gamma_\eta\left(\frac{\hat{\nu}}{2}\right)}\left|\frac{\hat{\Psi}}{1 + \tilde{z}_t^\top \hat{V}\tilde{z}_t}\right|^{-\frac{\eta}{2}}\left|I + \left(\tilde{x}_{t+1} - \hat{M}\tilde{z}_t\right)^\top \frac{\hat{\Psi}^{-1}}{1 + \tilde{z}_t^\top \hat{V}\tilde{z}_t}\left(\tilde{x}_{t+1} - \hat{M}\tilde{z}_t\right)\right|^{-\frac{\hat{\nu}+\eta}{2}}$$
$$= \mathcal{T}_{\hat{\nu}}\left(\hat{M}\tilde{z}_t, \Psi\left(1 + \tilde{z}_t^\top \hat{V}\tilde{z}_t\right)\right) \tag{26}$$

Which is a multivariate t-distribution with mean $\hat{M}\tilde{z}_t$ and scale matrix $\Psi\left(1 + \tilde{z}_t^\top \hat{V}\tilde{z}_t\right)$.

Under large number of degrees of freedom (large number of training examples in our case), this distributions converges to a multivariate gaussian distribution with mean $\hat{M}\tilde{z}_t$ and a covariance $\Psi\left(1 + \tilde{z}_t^\top \hat{V}\tilde{z}_t\right)$.

From this derivation, the posterior predictive distribution is a multivariate Student-t distribution with mean $\hat{M}\mathbf{z}_t$ and scale matrix $\Psi\left(1 + \mathbf{z}_t^\top \hat{V}\mathbf{z}_t\right)$. As the degrees of freedom ($\nu$) increase, this distribution approaches a multivariate Gaussian. The assumption 'N is large' refers to the degrees of freedom in the Student-t distribution, equal to the number of data points $N$. For $N \geq 30$, the Student-t distribution closely approximates a multivariate Gaussian, with higher accuracy for $N \geq 50$–$100$. As our dataset size is orders of magnitude larger than 100, this assumption is well justified.

## B    TRUCK AND TRAILER DATA COLLECTION

One of the key contributions of this work is the application of motion planning techniques for autonomous truck and trailer systems. Autonomous driving datasets are typically expensive to acquire and maintain, with access often restricted to select (OEMs) and their suppliers. This study involved extensive data collection from a a real autonomous truck and trailer platform to develop test datasets that accurately reflect the complex dynamics of these systems under diverse and challenging conditions, including harsh winter environments. Data was gathered over 12 months (March 2023 – March 2024) using a 37.5-ton, 17-meter-long Scania autonomous tractor-semitrailer in various driving and weather scenarios across Sweden, covering all seasonal variations (see Fig. 6). Each session was supervised by a safety driver and test engineer to ensure safety and system reliability.

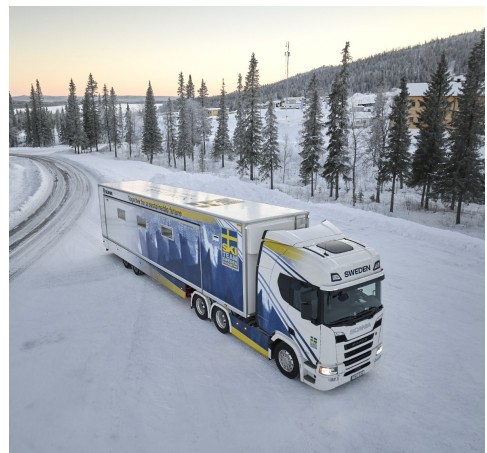

Figure 6: Truck and Semi-Trailer System. Image Courtesy of Scania CV AB.

The dataset includes comprehensive autonomy-related logs from various high fidelity sensors. Special emphasis was placed on capturing edge cases and distribution shifts, particularly under challenging winter conditions. Routine tests included maneuvers on a high-speed test track and test drives on public highways in different weather conditions including sun, snow and rain. Driving scenarios included straight line driving, negotiating curves and slopes, lane changes, cut-ins, stopping, following other actors and highway driving. Data was collected for fully autonomous driving, while more dangerous maneuvers were collected by manually driving the vehicle with sensors and logging enabled.

To assess autonomous performance in harsh winter conditions, specialized testing was conducted in February and March 2024 on dedicated tracks in northern Sweden. The vehicle underwent rigorous autonomous evaluations on packed snow and ice, navigating scenarios such as cornering on snow, responding to ice patches, executing sudden avoidance maneuvers, and performing sharp braking. Furthermore, test driving and sudden braking maneuvers were conducted on a mu-split track (see Fig. 7), where one side of the truck operated on dry asphalt while the other side navigated ice. This split-friction setup, known for inducing yaw moments, increases the risk of trailer swings and jackknifing. These tests were instrumental in analyzing vehicle stability and refining control strategies to mitigate such risks effectively.

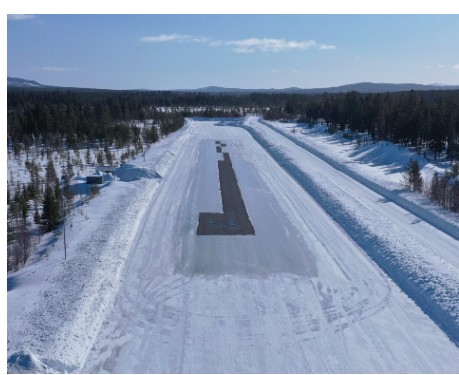

Figure 7: Mu-split test scenario. Image courtesy of Colmis AB.

After data collection, the dataset was filtered, normalized, and conditioned as outlined in Section 4. Long drives primarily consisted of straight-line driving interspersed with occasional steering and braking maneuvers. From the collected data, 10 hours of representative samples from various drives over the year were selected for training. The training dataset was curated to ensure diversity in maneuver types, capping straight-line driving at 40% and including critical scenarios. The test dataset, particularly winter test data, emphasized varied conditions such as road slopes, turns, and mu-split scenarios. This approach highlights challenging scenarios and distribution shifts, providing a rigorous evaluation of performance under diverse and difficult conditions.

# C  EXPERIMENTAL SETUP (SIMULATED ENVIRONMENTS)

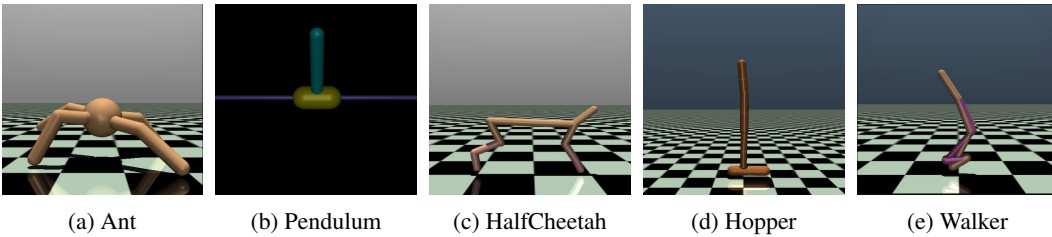

(a) Ant    (b) Pendulum    (c) HalfCheetah    (d) Hopper    (e) Walker

Figure 8: Environments: Ant, Inverted Pendulum, HalfCheetah, Hopper, Walker

## C.1  SIMULATION ENVIRONMENTS

### C.1.1  ANT - QUADRUPEDAL ROBOT

The **Ant** environment models a four-legged quadrupedal robot, based on OpenAI Gym Brockman et al. (2016). The primary objective is for the robot to advance forward as quickly as possible by learning to regulate the torques at its joints. Each of the robot's legs is equipped with two actuated joints, where the torque values are restricted within the range $[-1, 1]$. The aim is to achieve maximum forward velocity while maintaining stability on flat terrain. An illustration of the Ant environment is shown in Figure 8a.

### C.1.2  INVERTED PENDULUM

The **Inverted Pendulum** environment, adapted from OpenAI Gym Brockman et al. (2016), focuses on the challenge of balancing a pole mounted on a cart. The cart can move along a horizontal track, while the pole, hinged at its base, must remain upright. The action space consists of a continuous horizontal force applied to the cart, $a \in [-10, 10]$, and the goal is to prevent the pole from falling. If the pole's angle $\theta$ exceeds a threshold of $\pm 12°$, the episode terminates. Episodes are evaluated over 1000 time-steps, during which the system must maintain balance. An illustration of the Inverted Pendulum environment is shown in Figure 8b.

### C.1.3  HALFCHEETAH

The **HalfCheetah** environment simulates a two-legged running robot, modeled after locomotion tasks in OpenAI Gym Brockman et al. (2016). The robot's objective is to learn how to run forward by applying torques to its leg joints, with torque values constrained within the range $[-1, 1]$. The aim is to achieve maximum forward velocity while maintaining stability. Each episode spans 1000 steps, where performance is evaluated based on speed, with penalties for excessive energy usage. An illustration of the HalfCheetah environment is shown in Figure 8c.

### C.1.4  HOPPER

The **Hopper** environment, adapted from OpenAI Gym Brockman et al. (2016), simulates a one-legged robot tasked with learning to hop forward. The robot features three actuated joints—foot, knee, and hip—each controlled by continuous torque inputs within the range $[-1, 1]$. The goal is to achieve efficient forward movement while preventing the robot from falling. Episodes span 1000 timesteps, with performance evaluated based on rewards for forward motion and penalties for instability. An illustration of the Hopper environment is shown in Figure 8d.

### C.1.5  WALKER

The **Walker** environment simulates a bipedal robot, inspired by OpenAI Gym's locomotion tasks Brockman et al. (2016), where the robot must learn to walk forward. The robot has two legs, each equipped with actuated joints at the hip, knee, and ankle, with torques controlled by continuous values in the range $[-1, 1]$. The objective is to move forward while maintaining balance and avoiding

falls. Episodes last for 1000 timesteps, and the robot is rewarded for forward progress, with penalties applied for instability and excessive energy consumption. An illustration of the Walker environment is shown in Figure 8e.

### C.2    DATA COLLECTION USING TD3 AGENT

The data for these environments was collected using a trained expert agent, specifically a Twin Delayed Deep Deterministic (TD3) agent Fujimoto et al. (2018). The TD3 agent was trained for 1 million timesteps before being used for data collection. A total of 100k trajectories were collected for both training and testing, with 10k trajectories randomly selected from each set for the final dataset.

Each environment was run under three different conditions: (1) with process noise, (2) with observation noise, and (3) without noise (clean). The resulting datasets were stored for future use in training and evaluation purposes.

# D  ADDITIONAL RESULTS

The following section provides a detailed evaluation of the proposed methods, Blak and BlakVar, across five simulated environments: Hopper, HalfCheetah, Ant, InvertedPendulum, and Walker2d. Each environment is tested under three conditions: normal (no noise), observation noise, and process noise. These settings are designed to assess the models' robustness, adaptability, and overall performance in dynamic and complex control tasks. Results are presented in two forms: validation loss plots during the training process, which illustrate the learning dynamics over time, and consolidated tables showing the final loss (mean ± standard deviation) for each environment and condition.

Blak, which leverages a robust transformer-based architecture, and BlakVar, designed for real-time adaptability with a variational transformer decoder, are compared against a range of baseline methods, including MCDropout, BayesianNN, Dko, NODE, Desko, and EMLP. The following analysis highlights the strengths of Blak and BlakVar, particularly in handling noise and maintaining low loss across environments with varying levels of complexity.

## D.1  INVERTEDPENDULUM

The InvertedPendulum environment, due to its simplicity, serves as a baseline for evaluating model performance. The results are summarized in Table 3, and Figure 9 shows the loss/valid trends for each condition.

| Baseline | Normal | Observation Noise | Process Noise |
|---|---|---|---|
| MCDropout | $0.0093 \pm 0.0008$ | $0.0217 \pm 0.0031$ | $0.0175 \pm 0.0023$ |
| Blak | $0.0002 \pm 0.0000$ | $0.0009 \pm 0.0001$ | $0.0010 \pm 0.0000$ |
| BayesianNN | $0.0210 \pm 0.0030$ | $0.0182 \pm 0.0006$ | $0.0286 \pm 0.0034$ |
| Dko | $\mathbf{0.0001 \pm 0.0000}$ | $\mathbf{0.0008 \pm 0.0000}$ | $\mathbf{0.0009 \pm 0.0000}$ |
| NODE | $0.0033 \pm 0.0028$ | $0.0093 \pm 0.0021$ | $0.0058 \pm 0.0008$ |
| Desko | $0.0006 \pm 0.0001$ | $0.0012 \pm 0.0000$ | $0.0022 \pm 0.0000$ |
| EMLP | $\mathbf{0.0001 \pm 0.0000}$ | $0.0002 \pm 0.0009$ | $0.0012 \pm 0.0001$ |
| BlakVar | $0.0008 \pm 0.0004$ | $0.0012 \pm 0.0008$ | $0.0042 \pm 0.0010$ |

Table 3: Consolidated final loss (mean ± std) across noise conditions for InvertedPendulum. Best results are highlighted in **bold**.

All methods performed well in the InvertedPendulum environment due to its simplicity, as reflected in the uniformly low loss values. Blak demonstrated competitive performance across all conditions, achieving consistently low losses, such as 0.0002 under normal conditions and , 0.0009 under observation noise, and 0.0010 under process noise. BlakVar, optimized for real-time adaptability, maintained reasonable performance with losses of 0.0012 under observation noise and 0.0042 under process noise, despite its focus on generalization.

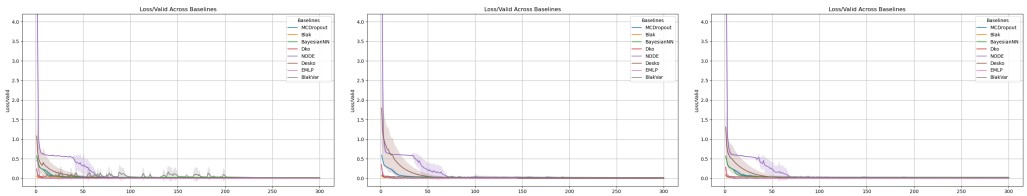

Figure 9: Loss/valid trends for the InvertedPendulum environment under different conditions. (Left) Normal. (Right) Observation Noise. (Center) Process Noise.

## D.2  WALKER2D

The Walker2d environment evaluates more complex dynamics compared to InvertedPendulum. The results, shown in Table 4, highlight that **Blak performs strongly across all noise conditions**, with **BlakVar** also showing competitive performance. Figure 10 visualizes the loss/valid trends.

| Baseline | Normal | Observation Noise | Process Noise |
|---|---|---|---|
| MCDropout | $0.2532 \pm 0.0220$ | $0.2936 \pm 0.0163$ | $2.2414 \pm 0.0266$ |
| Blak | $\mathbf{0.0368 \pm 0.0017}$ | $\mathbf{0.0596 \pm 0.0021}$ | $\mathbf{0.9453 \pm 0.0190}$ |
| BayesianNN | $0.2249 \pm 0.0097$ | $0.5475 \pm 0.0071$ | $1.9081 \pm 0.0444$ |
| Dko | $0.1694 \pm 0.1615$ | $0.2037 \pm 0.1678$ | $1.1754 \pm 0.3908$ |
| NODE | $0.1345 \pm 0.0067$ | $0.1596 \pm 0.0077$ | $1.3679 \pm 0.0630$ |
| Desko | $0.7040 \pm 0.7137$ | $0.4143 \pm 0.0203$ | $2.3168 \pm 0.0155$ |
| EMLP | $0.3110 \pm 0.0223$ | $0.3427 \pm 0.0368$ | $1.5425 \pm 0.0292$ |
| BlakVar | $0.1838 \pm 0.0104$ | $0.2098 \pm 0.0024$ | $1.0422 \pm 0.0036$ |

Table 4: Consolidated final loss (mean $\pm$ std) across noise conditions for Walker2d. Best results are highlighted in **bold**.

Blak achieves the lowest final loss across all noise conditions, with notable margins under normal conditions (0.0368) and observation noise (0.0596). BlakVar also performs well, achieving competitive results such as 0.1838 under normal conditions and 1.0422 under process noise. While Blak outperforms in most scenarios, the results of BlakVar demonstrate its adaptability and strength in handling complex environments, particularly in noisy settings.

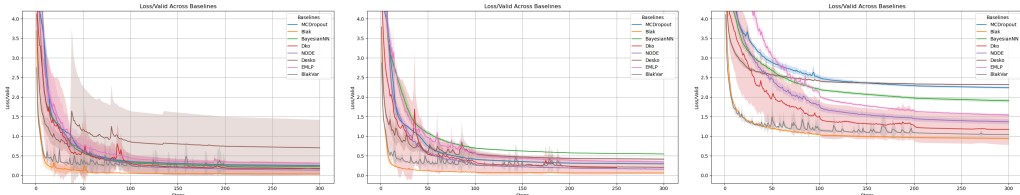

Figure 10: Loss/valid trends for the Walker2d environment under different conditions. (Left) Normal. (Center) Observation Noise. (Right) Process Noise.

A key observation is the overall robustness of both Blak and BlakVar under noise. While Blak consistently achieves lower losses, BlakVar remains competitive, especially in scenarios with process noise, where its final loss of 1.0422 is close to that of Blak. The gap between these methods and the other baselines increases with noise, emphasizing their capability to handle challenging conditions effectively.

## D.3 HOPPER

The Hopper environment introduces moderately complex dynamics, making it a good benchmark for evaluating robustness and adaptability. The results, summarized in Table 5, demonstrate that **Blak achieves the lowest loss across all conditions**, while **BlakVar** shows competitive performance, especially under noisy settings. Figure 11 illustrates the loss/valid trends across the three noise conditions.

| Baseline | Normal | Observation Noise | Process Noise |
|---|---|---|---|
| MCDropout | $0.3904 \pm 0.0056$ | $0.4813 \pm 0.0153$ | $0.8418 \pm 0.0217$ |
| Blak | $\mathbf{0.0478 \pm 0.0026}$ | $\mathbf{0.0916 \pm 0.0024}$ | $\mathbf{0.2581 \pm 0.0051}$ |
| BayesianNN | $0.2923 \pm 0.0121$ | $0.6607 \pm 0.0182$ | $0.6950 \pm 0.0113$ |
| Dko | $0.0911 \pm 0.0148$ | $0.1321 \pm 0.0158$ | $0.3318 \pm 0.0512$ |
| NODE | $0.0887 \pm 0.0146$ | $0.1326 \pm 0.0119$ | $0.2766 \pm 0.0182$ |
| Desko | $0.4193 \pm 0.0287$ | $0.5291 \pm 0.0089$ | $1.0950 \pm 0.0332$ |
| EMLP | $0.1823 \pm 0.0113$ | $0.2723 \pm 0.0345$ | $0.4921 \pm 0.0582$ |
| BlakVar | $0.1604 \pm 0.0128$ | $0.1938 \pm 0.0098$ | $0.3127 \pm 0.0196$ |

Table 5: Consolidated final loss (mean $\pm$ std) across noise conditions for Hopper. Best results are highlighted in **bold**.

Blak consistently achieves the lowest final loss under all conditions, with notable results such as 0.0478 under normal conditions and 0.0916 under observation noise. BlakVar, while slightly less performant, remains competitive with results such as 0.1604 under normal conditions and 0.3127 under process noise. Among the baselines, NODE and Dko show relatively strong results, although they consistently lag behind Blak.

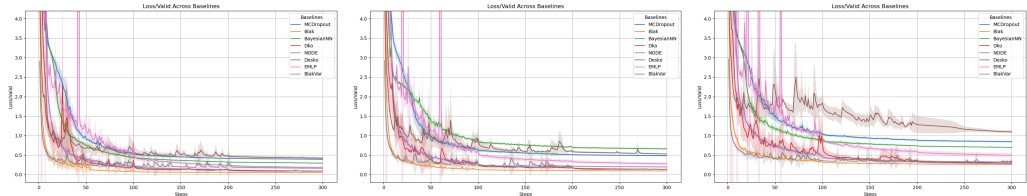

Figure 11: Loss/valid trends for the Hopper environment under different conditions. (Left) Normal. (Center) Observation Noise. (Right) Process Noise.

A key observation in this environment is that the gap between Blak and other methods widens as noise increases, particularly under process noise. BlakVar, designed for real-time adaptability, maintains strong performance and closes the gap to Blak in noisier conditions. Among the baselines, NODE and Dko demonstrate robust performance but fail to match the adaptability of Blak and BlakVar under higher noise levels.

### D.4 ANT

The Ant environment represents a highly complex control scenario, testing the robustness of models in handling intricate dynamics. The results, summarized in Table 6, show that **Blak performs consistently well across all conditions**, while **BlakVar** demonstrates competitive performance, particularly under noisier settings. Figure 12 visualizes the loss/valid trends for this environment.

| Baseline | Normal | Observation Noise | Process Noise |
|---|---|---|---|
| MCDropout | $0.4005 \pm 0.0038$ | $0.4225 \pm 0.0032$ | $0.6478 \pm 0.0045$ |
| Blak | $\mathbf{0.2646 \pm 0.0028}$ | $\mathbf{0.2846 \pm 0.0046}$ | $\mathbf{0.4143 \pm 0.0017}$ |
| BayesianNN | $0.4054 \pm 0.0132$ | $0.5756 \pm 0.0116$ | $0.6136 \pm 0.0108$ |
| Dko | $0.3340 \pm 0.0331$ | $0.3428 \pm 0.0367$ | $0.3255 \pm 0.0315$ |
| NODE | $0.6743 \pm 0.0493$ | $0.7446 \pm 0.0699$ | $0.7428 \pm 0.0189$ |
| Desko | $0.5318 \pm 0.0036$ | $0.5633 \pm 0.0029$ | $0.7746 \pm 0.0099$ |
| EMLP | $0.3159 \pm 0.0112$ | $x \pm x$ | $0.3966 \pm 0.0396$ |
| BlakVar | $0.3929 \pm 0.0263$ | $0.4721 \pm 0.0249$ | $0.6092 \pm 0.0246$ |

Table 6: Consolidated final loss (mean $\pm$ std) across noise conditions for Ant. Best results are highlighted in **bold**. An x denotes that the method was unable to converge

Blak achieves the lowest loss in all noise conditions, with results such as 0.2646 under normal conditions and 0.4143 under process noise. BlakVar maintains strong results, with losses of 0.3929 under normal conditions and 0.6092 under process noise, showing its adaptability to challenging environments. Among the baselines, Dko and EMLP exhibit robust results but remain less effective in high-noise scenarios.

A key observation in this environment is the stability of Blak's performance across all noise conditions, despite the complexity of the task. BlakVar, while not as consistent as Blak, remains competitive and demonstrates its strength under process noise. Among the baselines, NODE struggles with noise, while Dko and EMLP perform relatively well but show higher variability.

### D.5 HALFCHEETAH

The HalfCheetah environment, known for its challenging dynamics and complexity, serves as a benchmark for testing the resilience and adaptability of models. The results, shown in Table 7,

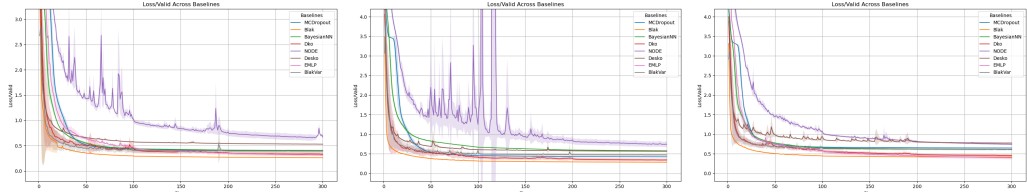

Figure 12: Loss/valid trends for the Ant environment under different conditions. (Left) Normal. (Center) Observation Noise. (Right) Process Noise.

| Baseline | Normal | Observation Noise | Process Noise |
|---|---|---|---|
| MCDropout | $3.8633 \pm 0.0444$ | $5.2776 \pm 0.2730$ | $4.3792 \pm 0.1295$ |
| Blak | $\mathbf{0.7480 \pm 0.0095}$ | $\mathbf{1.4726 \pm 0.0101}$ | $\mathbf{0.8866 \pm 0.0150}$ |
| BayesianNN | $1.7198 \pm 0.0461$ | $5.2428 \pm 0.1864$ | $2.1519 \pm 0.3580$ |
| Dko | $2.8472 \pm 2.3590$ | $2.7811 \pm 1.5627$ | $2.5508 \pm 2.0130$ |
| NODE | $4.8976 \pm 0.6158$ | $6.4289 \pm 0.5980$ | $6.3986 \pm 1.1861$ |
| Desko | $1.7308 \pm 0.1716$ | $3.4415 \pm 0.2946$ | $2.0143 \pm 0.1917$ |
| EMLP | $x \pm x$ | $x \pm x$ | $x \pm x$ |
| BlakVar | $1.1722 \pm 0.0192$ | $1.7671 \pm 0.1189$ | $1.3070 \pm 0.0140$ |

Table 7: Loss/valid trends for the Half-Cheetah environment under different conditions. (Left) Normal. (Center) Observation Noise. (Right) Process Noise.

indicate that both **Blak and BlakVar exhibit strong performance across all conditions**, with Blak achieving the best results. Figure 13 illustrates the loss/valid trends.

Blak achieves the lowest loss in all conditions, demonstrating its ability to adapt to the environment's complexity. Notably, it records a final loss of 0.7480 under normal conditions, 1.4726 under observation noise, and 0.8866 under process noise. BlakVar also shows strong performance, with results such as 1.1722 under normal conditions and 1.3070 under process noise, indicating its robustness and adaptability to challenging conditions.

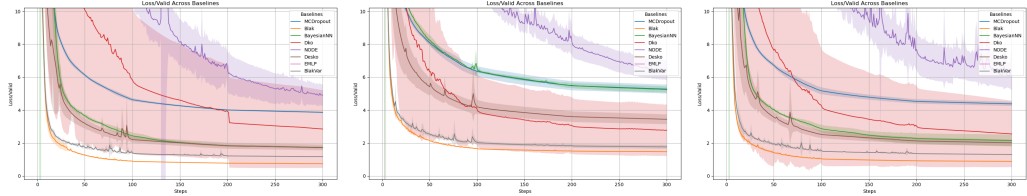

Figure 13: Loss/valid trends for the HalfCheetah environment under different conditions. (Left) Normal. (Center) Observation Noise. (Right) Process Noise.

One key observation is the large performance gap between Blak and the baselines, especially under noisy conditions. The results highlight Blak's ability to maintain low losses even in complex scenarios, while BlakVar remains competitive, particularly under process noise. Baselines like Dko and NODE struggle with the complexity of this environment, showing higher variability and loss values.

### D.6 MOTION PLANNING IMPLEMENTATION

To address the motion planning problem for our truck and trailer system, we employ a sampling-based Rapidly-exploring Random Tree (RRT) algorithm. Unlike traditional RRT approaches that sample directly from the state space, we sample actions from the action embedding space thanks to the variational action encoder. The variational encoder maps the action space into a Gaussian latent space, enabling efficient sampling that accounts for the dynamics of the system. Each sampled action is then passed through the Bayesian Koopman dynamics model to predict the corresponding future states over a specified planning horizon. This approach ensures that the generated paths

are dynamically feasible and consistent with the complex physical constraints of the truck-trailer system.

The planner generates a tree of candidate trajectories by iteratively sampling actions, predicting state sequences, and evaluating the cost associated with each trajectory. At each iteration, the algorithm selects the trajectory that minimizes a cost function considering for example: proximity to the goal, smoothness, and collision avoidance. By integrating the Koopman-based dynamics model, the planner accurately predicts the tractor's positions, velocities, and angles, as well as the tractor-trailer articulation angle for the receding horizon lengths (10 seconds into the future in our case). This ensures that the planned paths navigate toward the goal while avoiding obstacles and maintaining dynamic feasibility. Algorithm 1 outlines the proposed planning approach. To validate our planning algorithm we apply it to a motion planning problem that involves navigating a truck and trailer system towards randomly generated goal states while ensuring dynamic feasibility and obstacle avoidance. The goal states are randomly spawned within a predefined region of the planning space, representing potential destinations the vehicle must reach. At each time step, the planner is tasked with determining the positions, velocities, and angular configurations of the tractor, as well as the articulation angle between the tractor and trailer, over a specified planning horizon. The planner must not only compute a collision-free path to the goal but also ensure that the generated trajectory respects the physical and dynamic constraints of the truck-trailer system. This requires precise handling of the vehicle's nonlinear dynamics, particularly in complex scenarios involving tight turns, steep gradients, or low-traction surfaces. An example is illustrated in Fig. 5

---

**Algorithm 1** RRT-based Motion Planning Using Variational Action Sampling

---

**Require:** Goal state $s_{\text{goal}}$, initial state $s_{\text{start}}$, planning horizon $H$, number of iterations $N$
**Ensure:** Dynamically feasible path from $s_{\text{start}}$ to $s_{\text{goal}}$
1: Initialize tree $T$ with root node $s_{\text{start}}$
2: **for** $i = 1$ to $N$ **do**
3:     Sample action $a \sim \mathcal{N}(\mu, \sigma^2)$ from the variational action encoder
4:     Predict future states $s_t, s_{t+1}, \ldots, s_{t+H}$ using Koopman dynamics model
5:     **if** predicted trajectory reaches goal and avoids obstacles **then**
6:         Add predicted trajectory to tree $T$
7:     **end if**
8: **end for**
9: Select optimal trajectory from $T$ that minimizes cost function $J$, considering:
- Distance to goal: $\|s_H - s_{\text{goal}}\|$
- Smoothness: $\sum_{t=1}^{H} \|a_t - a_{t-1}\|$
- Collision avoidance: penalty for states near obstacles

10: **return** optimal trajectory

---

### D.7 RUNTIME ANALYSIS

All runtime experiments were conducted on an NVIDIA A10 GPU to ensure consistency and comparability. The reported runtimes correspond to the inference phase on the Hopper environment, chosen for its moderate complexity and suitability for benchmarking. Each method was evaluated based on its average runtime per inference step.

For our method, BLAK, the runtime includes the complete inference process using the Bayesian Koopman operator. For BLAKVar, the runtime reflects the efficiency of directly sampling from the variational embedding space, bypassing the action encoding step. This modification significantly improves computational efficiency while maintaining prediction accuracy. The runtimes for other methods are measured under standard inference settings. All reported times represent the mean of 1,000 inference operations to ensure reliability and eliminate variability due to hardware fluctuations.

Table 8: Inference Runtime Analysis (Average Time per Step in milliseconds).

| Method | Runtime (ms) |
|---|---|
| BLAK | .135 |
| BLAKVar | .097 |
| Deep Koopman Operator (DKO) | .206 |
| Deep Stochastic Koopman (Desko) | .110 |
| Bayesian Neural Networks (BNNs) | 1.35 |
| MC Dropout | .947 |
| Neural ODEs (NODE) | .209 |
| Ensemble MLP (EMLP) | .494 |

# E  ABLATION STUDIES

In this section, we conduct ablation studies to analyze the impact of key components in our method. Specifically, we evaluate the contribution of Bayesian learning across all environments and analyze the effect of varying history lengths on prediction accuracy in both noise-free and noisy settings. These experiments help identify configurations that offer the best trade-off between accuracy and computational efficiency.

## E.1  EFFECT OF BAYESIAN LEARNING

We compare the performance of our method with and without Bayesian learning across all six environments, including both noise-free and noisy conditions. Table 9 summarizes the results, showcasing the importance of Bayesian modeling in improving prediction accuracy and robustness across diverse scenarios.

Table 9: Performance with and without Bayesian learning (Mean Squared Error ± Standard Deviation).

| Environment | Without Bayesian Learning | With Bayesian Learning |
|---|---|---|
| Truck Dynamics | 0.1306 | **0.1016** |
| Hopper | 0.0482 | **0.0478** |
| HalfCheetah | 0.3064 | **0.2646** |
| Ant | 0.2131 | **0.2002** |
| Walker2d | 0.0383 | **0.0368** |
| InvertedPendulum | 0.0042 | **0.0002** |

## E.2  EFFECT OF HISTORY LENGTH

To evaluate the impact of history length ($q$), we conduct experiments on four environments: two noise-free settings (Hopper, HalfCheetah) and two noisy settings (Truck Dynamics, Ant with Process Noise). The history length determines the amount of temporal context the trajectory encoder incorporates, which can influence prediction accuracy. Table 10 reports the results for varying history lengths, highlighting the trade-offs between context size and accuracy.

Table 10: Effect of History Length on Prediction Accuracy (Mean Squared Error ± Standard Deviation).

| Environment | 1 | 5 | 10 | 20 |
|---|---|---|---|---|
| Hopper (Noise-Free) | 0.0516 | 0.0509 | 0.0493 | **0.0478** |
| HalfCheetah (Noise-Free) | 0.8223 | 0.7952 | 0.7624 | **0.7480** |
| Truck Dynamics (Noisy) | 0.1142 | 0.1101 | 0.1060 | **0.1016** |
| Ant (Process Noise) | 0.4633 | 0.4521 | 0.4349 | **0.4143** |

## F  IMPLEMENTATION DETAILS

This section provides comprehensive implementation details for the proposed method, Bayesian Learning with Adaptive Koopman Operators (BLAK), as well as concise descriptions of the baselines used for comparison. We detail the architectural components and training hyperparameters employed in the experiments. Common configurations across different experiments are grouped at the end to avoid repetition.

### F.1  PROPOSED METHOD: BAYESIAN LEARNING WITH ADAPTIVE KOOPMAN OPERATORS (BLAK)

Our proposed method integrates Koopman operator theory with transformer-based architectures to model system dynamics effectively. The implementation leverages Bayesian learning for uncertainty quantification and adaptation to dynamic environments. We develop two architectures: *Blak* and *BlakVar*, both leveraging a Transformer-based encoder-decoder framework to model state-action dynamics. Blak employs deterministic decoder, while BlakVar extends this with variational decoder for real-time opreation. Both methods use the same transformer-based encoder-decoder architecture. Inputs are projected into the embedding dimension through linear layers. To effectively capture position-dependent information, we employ Rotary Positional Encoding (RoPE) (Su et al., 2024) to embed sequence positions. The encoder leverages regular self-attention mechanism. However, for the decoder, we employ sliding-window causal attention with a window size of 8, restricting attention to current and past context and ensuring causality in predictions. The encoder processes historical state-action sequences of 20 time steps as concatenated state-action vectors of dimension $n_x + n_a$, where $n_x$ and $n_a$ represent the state and action dimensions, respectively. A linear embedding layer projects these inputs into a hidden dimension of 32. The decoder handles future action sequences, which are similarly projected into the same hidden dimension using a linear embedding layer. A learnable start token is prepended to the decoder input sequence to initialize decoding.

### F.2  BASELINES IMPLEMENTATION DETAILS

Below, we provide concise implementation details for each baseline method. Common training configurations are summarized in Section F.3.

**Deep Koopman with Control (DKO)**  The DKO method models and controls nonlinear systems using two encoder networks—for state and action embeddings—and the Koopman operator. The state encoder maps input states to a higher-dimensional embedding concatenated with the original state, using the following fully connected layers: [State Dimension $\rightarrow 32 \rightarrow 64 \rightarrow 128 \rightarrow 64 \rightarrow 32 \rightarrow$ State Embedding Dimension] with ReLU activations (except output). The action encoder processes state-control interactions through the layers: [State+Control Dimension $\rightarrow 32 \rightarrow 64 \rightarrow 128 \rightarrow 64 \rightarrow 32 \rightarrow$ Control Embedding Dimension] with similar activations. The Koopman operator comprises a linear operator $\mathbf{A}$ (initialized Gaussian and orthogonalized via SVD) and a control matrix $\mathbf{B}$.

**Neural Ordinary Differential Equation (Neural ODE)**  The Neural ODE model captures continuous-time dynamics by modeling the time derivative of the state as a neural network function, integrated over time using an ODE solver. The ODE function $f_\theta(x, a)$ is parameterized by a neural network. The input layer consists of concatenated state and action, dimension $n_x + n_a$, followed by a fully connected layer with 32 neurons. Hidden layers have sizes 32, 64, 96, 128, 96, 64, 32 neurons with ReLU activations. The output layer outputs the time derivative of the state, dimension $n_x$. We use a fourth order Runge-Kutta method as an ODE solver implemented using the `torchdiffeq` library.

**Monte Carlo Dropout (MC Dropout).**  The MC Dropout model estimates predictive means and uncertainties by using dropout layers during training and inference. The dynamics function, parameterized by a neural network, takes a concatenated state-action input ($n_x + n_a$), passes through fully connected layers with sizes [$32 \rightarrow 32 \rightarrow 64 \rightarrow 96 \rightarrow 128 \rightarrow 96 \rightarrow 64 \rightarrow 32$], uses ReLU activations, a dropout rate of 0.15, and outputs the predicted next state ($n_x$).

**Ensemble Neural Networks.** The Ensemble Neural Network model uses ten independently trained feedforward networks to capture dynamics and estimate uncertainties via ensemble variance. Each network processes a concatenated state-action input ($n_x + n_a$) through fully connected layers [$32 \rightarrow 32 \rightarrow 64 \rightarrow 96 \rightarrow 64 \rightarrow 32$] with ReLU activations and outputs the predicted next state ($n_x$).

**Bayesian Neural Network (BNN)** The Bayesian Neural Network (BNN) models predictive uncertainty by treating weights as Gaussian distributions. The dynamics model uses Bayesian linear layers with variational distributions over weights and biases. Input state-action vectors ($n_x + n_a$) pass through layers with dimensions: [$32 \rightarrow 64 \rightarrow 128 \rightarrow 64 \rightarrow 32$], ReLU activations, and output the next state ($n_x$). Priors are Gaussian with mean 0 and variance 1.

The loss combines mean squared error (MSE) and KL divergence between posterior and prior distributions:

$$\mathcal{L} = \mathcal{L}_{\text{data}} + \beta \, \text{KL}(\text{posterior} \parallel \text{prior}),$$

where $\beta = 10^{-5}$. During inference, Monte Carlo sampling provides mean predictions and variances, constructing diagonal covariance matrices for uncertainty estimation.

### F.3 COMMON IMPLEMENTATION DETAILS

Certain implementation aspects are common across all models, summarized here to avoid repetition.

All models (unless otherwise stated) are trained using the AdamW optimizer with a learning rate of $3 \times 10^{-3}$ and a weight decay of $1 \times 10^{-2}$. A `StepLR` scheduler is employed to reduce the learning rate by a factor of 0.3 at every one-third of the total training epochs. Training is conducted over 300 epochs with a batch size of 1024. The training horizon is set to 200 timesteps, and the prediction horizon extends to 300 timesteps. Input states and actions are standardized to have zero mean and unit variance.

The Mean Squared Error (MSE) between predicted and actual future states serves as the primary loss function. For *BlakVar* and Bayesian neural networks, an additional KL divergence loss is incorporated alongside the MSE loss. To ensure training stability, gradient clipping with a maximum norm of 1.0 is applied to all feedforward networks.

