# OpenReview forum: "Bayesian Learning of Adaptive Koopman Operator with Application to Robust Motion Planning for Autonomous Trucks"
_ICLR.cc/2025/Conference — Submitted to ICLR 2025_

### Official Review · Reviewer_kF8W · 2024-10-31

**Soundness:** 2
**Presentation:** 2
**Contribution:** 3
**Rating:** 5
**Confidence:** 3

**Summary:**

This paper considers the uncertainty and temporal distributional shift issues in the Koopman operator framework, and proposes to incorporate Bayesian learning to form adaptive Koopman operators. Experiment results on predicting truck dynamics and motion planning are shown to prove effectiveness.

**Strengths:**

The paper is in general clearly written, and the idea of incorporating Bayesian learning into adaptive Koopman operators seems novel. The authors also demonstrate improved results for state prediction and motion planning under uncertainties compared to several solid baseline methods.

**Weaknesses:**

While the idea of the paper seems novel, several key details are missing.

1. The pipeline of the proposed method does not seem clear. For instance, In Fig. 1, how is the embedding from the trajectory encoder combined with the embedding from the action encoder? How does the total loss for training look like given equation (6)?

2. Several assumptions in the theoretical part need justification, for example, in Lemma 3.2, “under the assumption that the number of datapoints N is large”, how is large defined and in practice can this condition be satisfied?

3. The experiment description is not very comprehensive. How are the baseline methods implemented? What is the goal and setting of the motion planning problem? Why were 200 and 300 chosen to be the epoch numbers? None of such information is presented in the main paper or the appendix.

A few typos in the paper:
1. ‘To mitigate these challenges, To address these challenges’ are repetitive in the Distribution Shift paragraph on page 3.
2. z_t should be \tilde z_t on page 4 before equation (3)?
3. In Table 2, BLAST is not consistent with BLAK that is used elsewhere for the proposed method?

**Questions:**

1. Why is \mathcal{K} of dimension \eta \times d?

2. Any comparison with adaptive Koopman operator methods? For example, the papers below.
https://arxiv.org/pdf/2202.09501
https://arxiv.org/pdf/2211.09512

3. How do you deal with error from finite dimensional approximation? This following paper considers it.
https://arxiv.org/pdf/2410.00703

---

> ### Author Response · Authors · 2024-11-24
> **Response to The Reviewer's Comments on Weaknesses 1/2**
>
> We thank the reviewer for the insightful and constructive feedback. We address each concern below:
>
> > **Weakness 1:** *The pipeline of the proposed method does not seem clear. For instance, In Fig. 1, how is the embedding from the trajectory encoder combined with the embedding from the action encoder? How does the total loss for training look like given equation (6)?*
>
> **Answer:** Here, we break the reviewer's concern into multiple parts.
>
> > 1. "*The pipeline of the proposed method does not seem clear* For instance, In Fig. 1, how is the embedding from the trajectory encoder combined with the embedding from the action encoder?".
>
> Our architecture consists of the following steps:
> - Step 1: we pass the past history of state-action pairs through a transformer encoder, which we call *trajectory encoder*, to obtain the embeddings for each time step of the history.
> - Step 2 (only in non-variational *action encoder*): we also embed future actions using a transformer decoder, which we call *action encoder*, in order to get future action embeddings.
> - Step 3: we use the past state-action embedding obtained from step 1, except the current (last) state-action embedding, to update our probabilistic Koopman operator online to address distribution shifts.
> - Step 4: using the current state-action pair which is obtained from step 1, the future embeddings vector obtained from step 2, and the updated beleif over the Koopman operator obtained from step 3, we obtain future states which are used later in motion planning. More specifically, this works as follows:
>     - Step 4.1: Obtain the embedding of the current timestep state-action pair $\tilde{z}$. This is the current timestep embedding which is obtained from the encoder (step 1.)
>     - Step 4.2: Use (Equation 14) in the paper to obtain the embedding of the state of the next timestep $\tilde{x}_{t+1}$
>     - Step 4.3: Concatenate the state embedding of time $t+1$, obtained from step 4.2, and the action embedding of time $t+1$, obtained from step 2, to get the embedding of the state-action pair of time $t+1$ which is $\tilde{z}_{t+1} = [\tilde{x}_{t+1}, \tilde{u}_{t+1}]$.
> - Repeat steps 4.1-4.3 for all future time steps using Equation 15 instead of 14.
> - It's worth emphasizing in the case of *variational action encoder*, we sample the embedded space directly (i.e.: sample from a standard gaussian) without passing the future actions through the *action encoder*. This allows real-time of our framework.
>
> > 2. "*How does the total loss for training look like given equation (6)?*"
>
> In the next revised verion, we will include visualizations for both the loss plots of our method as well as other baselines as well as visualizations for open-loop predictions.
>
> ---
>
> > **Weakness 2:** *Several assumptions in the theoretical part need justification, for example, in Lemma 3.2, “under the assumption that the number of datapoints N is large”, how is large defined and in practice can this condition be satisfied?*
>
> **Answer**: The assumption “N is large” refers to the degrees of freedom in the Student-t distribution. In our case, N denotes the number of data points, meaning that the number of degrees of freedom of the Student-t distribution is equal to the number of data points in our dataset. As $N$ increases, the Student-t distribution becomes increasingly similar to a multivariate normal distribution. Studies suggest that when $N$ is around 30 or higher, the approximation to the normal distribution becomes reasonably accurate. Degrees of freedom in the range of 50 to 100 or more are typically preferable [1]. Nevertheless, the number of datapoints in our case is orders of magnitude higher than 100, which makes our assumption here reasonable.

---

> ### Author Response · Authors · 2024-11-24
> **Response to The Reviewer's Comments on Weaknesses 2/2**
>
> > **Weakness 3:** *The experiment description is not very comprehensive. How are the baseline methods implemented? What is the goal and setting of the motion planning problem? Why were 200 and 300 chosen to be the epoch numbers? None of such information is presented in the main paper or the appendix.*
>
> **Answer:** We address the reviewer's concern part by part.
>
> - *The experiment description is not very comprehensive. How are the baseline methods implemented?*
>
> We have added a detailed implementation section in the appendix, which provides comprehensive information about our method and the baseline methods. This includes specifics such as parameters, embedding dimensions, and other relevant details.
> - *What is the goal and setting of the motion planning problem?*
>
> Our planning problem involves a standard path planning task, with the key requirement that the generated path must be dynamically feasible for our truck and trailer system. Specifically, the planner's objective is to generate a sequence of states over time to reach a randomly spawned goal. At each time step, the planner must determine the positions, velocities, and angles of the tractor, as well as the angle between the tractor and the trailer, for the entire planning horizon. The generated plan must navigate towards the goal while avoiding obstacles on the road. We have further detailed this in both the main text and the appendix.
>
> - *Why were 200 and 300 chosen to be the epoch numbers?*
>
> It's worth clarifying that 200 and 300 are the number of time steps of our predictions. Our method focuses on long horizon planning, unlike other methods that focus more on control [2, 3]. The number 200 originates from the fact that our planner is run at 20 Hz, and the planning horizons is 10 seconds. This makes 200 time stepsin total. Moreover, to confirm the generalization of our method outside the training sequence length, we evaluated against 300 time steps as well to make sure that our method indeed learns the correct dynamics. As for the epochs number, 300 was chosen as it was found to be enough for the baselines to converge. This has been clarified in the text.
>
> - *None of such information is presented in the main paper or the appendix.*
>
> We hope that this address the reviewer's concerns. Additional details are also found in the appendix.
>
> ---
>
> > **Weakness 4:** *A few typos in the paper:*
> > - *‘To mitigate these challenges, To address these challenges’ are repetitive in the Distribution Shift paragraph on page 3.*
> > - *$z_t$ should be $\tilde{z}_t$ on page 4 before equation (3)?*
> > - *In Table 2, BLAST is not consistent with BLAK that is used elsewhere for the proposed method?*
>
> **Answer:** We appreciate the reviewer’s feedback and have made the necessary corrections in the updated version of the manuscript.
>
>
> ---
>
> References:
>
> - [1] https://en.wikipedia.org/wiki/Student%27s_t-distribution
> - [2] Han, Minghao, Jacob Euler-Rolle, and Robert K. Katzschmann. "DeSKO: Stability-assured robust control with a deep stochastic Koopman operator." International Conference on Learning Representations. 2021.
> - [3] Shi, Haojie, and Max Q-H. Meng. "Deep Koopman operator with control for nonlinear systems." IEEE Robotics and Automation Letters 7.3 (2022): 7700-7707.

---

> ### Author Response · Authors · 2024-11-24
> **Response to The Reviewer's Questions**
>
> We thank the reviewer for the raising up these important questions. We answer them as follows:
>
> > **Question 1:** Why is $\mathcal{K}$ of dimension $\eta \times d$?
>
> **Answer:** The finite dimensional approximation of the Koopman operator, $\mathcal{K}$, acts of the complete embedding vector $\tilde{z}_t \in \mathbb{R}^d$
>
> to produce the embedding of the state at the next time step $\tilde{x}_{t+1} \in \mathbb{R}^\eta$.
>
> More specifically, $\tilde{x}_{t+1} = \mathcal{K} \tilde{z}_t$. Thus, $\mathcal{K}$ is of dimension $\eta \times d$.
>
> ---
>
> > **Question 2:** Any comparison with adaptive Koopman operator methods? For example, the papers below.  *https://arxiv.org/pdf/2202.09501*,  *https://arxiv.org/pdf/2211.09512*
>
> **Answer:** In [1], the observables are predefined functions chosen to lift the system’s state into a higher-dimensional space for linear approximation, rather than being learned or optimized. This approach uses a discrete-time formulation of the Koopman operator, approximated via Extended Dynamic Mode Decomposition (EDMD). Real-time adaptation is implemented through a recursive extension of EDMD (rEDMD), which employs a variable forgetting factor and stability measures to update the model dynamically as new data becomes available. This allows for adaptive control and observation, ensuring consistent performance even as system dynamics change.
>
> In contrast, [2] uses a spectral-collocation method to obtain observables, approximating them as polynomials evaluated at Gauss-Lobatto points. Their approach discretizes the Koopman operator spatially rather than temporally, creating a continuous embedding framework that leverages eigen-decomposition to approximate Koopman eigenvalues and eigenfunctions. Adaptation is achieved by dynamically updating these approximations using checkpoints and adjusting neighborhoods as needed to maintain accuracy. The Adaptive Spectral Koopman (ASK) method thus benefits from efficient eigensolvers and a mesh-free design, offering flexible and precise solutions for nonlinear ODEs.
>
> We believe these methods are more aligned with classical approaches, which is evident in their evaluation settings. Applying them to our systems would be challenging, as our scenarios assume unknown and very complex physical models—particularly in cases like modeling a truck on icy surfaces, where good nominal models are difficult to establish. Consequently, we leave a direct comparison with these methods to future work. Nevertheless, we thank the reviewer for highlighting these interesting studies.
>
> ---
>
> > **Question 3:** How do you deal with error from finite dimensional approximation? This following paper considers it. *https://arxiv.org/pdf/2410.00703*
>
> **Answer:** For [3], their approach combines Bayesian learning with Kalman smoothing within the Koopman framework to separate measurement noise from the data and iteratively refine the Koopman operator using the Expectation-Maximization (EM) algorithm. The observables are constructed using time-delay embedding, where the original state variables are augmented with their time-shifted versions to create an extended state space that better captures the system's dynamics. Notably, their method focuses on autonomous systems and does not account for control inputs, limiting its applicability to systems without external control. They handle finite-dimensional approximation errors by treating them as zero-mean Gaussian random variables, employing time-delay functions to approximate Koopman invariance, and applying Kalman smoothing to mitigate the impact of these errors on state estimation.
>
> In contrast, our method addresses finite-dimensional approximation errors differently. We employ Bayesian learning to model and quantify uncertainty directly within the Koopman operator and observation matrix. Specifically, we represent the Koopman operator as a probability distribution rather than a fixed entity, allowing us to capture and reason probabilistically about errors that arise from approximating the infinite-dimensional Koopman operator in a finite-dimensional space. By incorporating noise-adaptive mechanisms and using a Bayesian framework, we can account for both model uncertainty and observation noise. This probabilistic treatment provides robustness to approximation errors, enabling our method to adapt dynamically and deliver more reliable predictions even when the finite-dimensional representation introduces inevitable inaccuracies.
>
> ---
>
> Refernces:
>
> - [1] Junker, Annika, et al. "Adaptive Koopman-Based Models for Holistic Controller and Observer Design⋆." IFAC-PapersOnLine 56.3 (2023): 625-630.
> - [2] Li, Bian, et al. "The adaptive spectral Koopman method for dynamical systems." SIAM Journal on Applied Dynamical Systems 22.3 (2023): 1523-1551.
> - [3] Zeng, Zhexuan, et al. "Koopman Spectral Analysis from Noisy Measurements based on Bayesian Learning and Kalman Smoothing." arXiv preprint arXiv:2410.00703 (2024).

---

> > ### Comment · Reviewer_kF8W · 2024-11-27
> >
> > Thank you for answering my questions. Some of my concerns are addressed. However, I do not see any update on the paper.

---

> ### Author Response · Authors · 2024-11-30
> **Key Revisions and Enhancements**
>
> We sincerely thank the reviewer for their insightful feedback. In the revised paper, we have made the following key updates and enhancements:
>
> ---
>
> ## Key Revisions Addressing the Reviewer's Concerns:
>
> - **Improved Figure 1**:
>   We enhanced **Figure 1** by explicitly outlining the steps of our algorithm and introducing a key partition to distinguish different symbols. Additionally, the figure caption now clarifies how the embeddings are combined and provides a detailed explanation of the algorithmic steps.
>
> - **Visualization Enhancements**:
>   Visualizations of loss functions across various baselines and environments have been added to both the main experimental section and **Appendix D**.
>
> - **Clarification on Assumptions**:
>   We clarified the assumption that **N is large**, providing a reasonable range and discussing its practical relevance in our case. These details have also been addressed in the proof on page 13 of **Appendix A**.
>
> - **Implementation Details**:
>   **Appendix F** now includes comprehensive implementation details for our method and the baselines, along with the specific parameter values used for each algorithm.
>
> - **Experimental Setup Updates**:
>   We clarified the reason behind the use of 200 and 300 steps for prediction in the **Experimental Results** section (page 9) and explained the rationale behind selecting 300 epochs in the same paragraph.
>
> - **Motion Planning Section**:
>   A dedicated section on motion planning has been added, accompanied by a pseudo-algorithm for generating dynamically feasible paths using our model. These updates are included in **Appendix D (Additional Results)** on pages 23–24.
>
> ---
>
> ## Additional Enhancements:
>
> - **Expanded Baseline Comparisons**:
>   We incorporated comparisons with additional baselines, including methods based on Koopman theory (e.g., Neural ODEs) and uncertainty quantification techniques (e.g., Bayesian NN and MC Dropouts). These results are presented in the **Experimental Setup** section and **Appendix D**.
>
> - **Computational Evaluation**:
>   A computational assessment has been conducted and is now detailed in **Appendix D** (pages 24–25).
>
> - **Real Vehicle Data Collection**:
>   **Appendix B** includes an in-depth explanation of the data acquisition and processing methods used in the study.
>
> - **Robustness Testing**:
>   We enhanced the test dataset for truck dynamics by including winter test data with aggressive maneuvers absent from the training data. These updates, demonstrating robustness in out-of-distribution scenarios (e.g., braking in a μ-split scenario), are detailed in the **Experimental Setup** section (pages 7–10) and **Appendix B** (page 17).
>
> - **Ablation Studies**:
>   Ablation studies have been conducted to evaluate the benefits of Bayesian learning and the impact of sequence length on performance. These results are presented across varying sequence lengths. This can be found in Appendix E.
>
> ---
>
> ## Supplementary Materials:
>
> - **Revised Manuscript**:
>   A revised manuscript with all updates clearly highlighted in blue.
>
> - **Open-Source Code**:
>   Open-source code, including the data collection process and model training procedures for simulated environments.
>
> ---
>
> We deeply appreciate the reviewer’s valuable comments. We believe these revisions comprehensively address the concerns raised, providing a more robust and detailed evaluation of our proposed method.
>
> We hope these updates, along with our prior responses, encourage the reviewer to re-evaluate our submission positively. We remain fully open to any additional feedback or questions the reviewer may have.

---

### Official Review · Reviewer_RNwV · 2024-11-02

**Soundness:** 3
**Presentation:** 3
**Contribution:** 3
**Rating:** 6
**Confidence:** 3

**Summary:**

This paper proposes a Bayesian Koopman operator for modeling of dynamical systems, that incorporates uncertainty quantifica-
tion. The goal is to make it flexible enough to deal with distributional shifts, which is achieved with an online adaptation mechanism, ensuring the operator remains responsive to changes in system dynamics. Distributional shifts are detected with a specific change variable.

The approach is applied to motion planning and evaluated via a dataset of real-world truck dynamics data under varying weather conditions, and on other simulated environments. The original approach (using a transformer-based encoder), and a variant optimized for computational efficiency (variational encoder) are compared to several other Koopman operator-based approaches incorporating uncertainty quantification. The proposed approach shows best performance in the simulations, when it is not reduced for computational performance.

**Strengths:**

The proposed Koopman-based approach leveraging Bayesian learning for dynamic systems and distributional shifts in such systems is original. The paper is well written with clear presentation. Extending Koopman-operator based modeling to adapt to ucertainties is solving an improtant problem in plannig and control.

The evaluation of the method is providing a comparison with state of the art methods in the field. A realistic dataset and some other simulated environments are used to benchmark the method.

**Weaknesses:**

1) I believe there is a piece of information which needs to be added, in particular computational times of the proposed approach. It would be interesting to directly compare the gain in computational efficiency from applying variational encoder. A table showing the corresponding computational times will help understanding the potential of the approach for realistic application. Especially for path planning, the assumptions, the time windows (how long is the sequence), the values of the tempering parameter, and the computational time will help demonstrating the efficiency of the algorithm.

2) Some justification in picking the transformer-based encoder will be helpful. Is there a real benefit of using it, given the overhead and the large amount of data needed to train it? Why using it, if afterwards they have to be reduced to variational encoders with lower performance, and with tempering parameters?

3) It would be useful to see a comparion in the path planning comparison with an approach which is not based on Koopman-representation, such as hierarchical planning (A* and RRT) or policy optimization, or any other approach.


Minor:
L.145 - repetition
L.200 - adopt --> adopt

**Questions:**

Under the assumption that the noise vectors are i.i.d sampled from a multivariate Gaussian distribution, the learning of the Koopman operator using Bayesian LR model proceeds.
- How important is this assumption? Is it valid realistically for autonomous driving?

---

> ### Author Response · Authors · 2024-11-24
> **Response to The Reviewer's Comments on Weaknesses 1/3**
>
> > **Weakness 1:** *I believe there is a piece of information which needs to be added, in particular computational times of the proposed approach. It would be interesting to directly compare the gain in computational efficiency from applying variational encoder. A table showing the corresponding computational times will help understanding the potential of the approach for realistic application. Especially for path planning, the assumptions, the time windows (how long is the sequence), the values of the tempering parameter, and the computational time will help demonstrating the efficiency of the algorithm.*
>
> **Answer:** We thank the reviewer for the insightful suggestion. To address the concern about computational efficiency, we will add a new section in the appendix that compares the computational times of our approach—both with and without the variational encoder—alongside other baseline methods. This table highlights the efficiency gains achieved with the variational encoder, especially relevant for real-time applications.
>
> For path planning, we have expanded on the experimental setup and assumptions, specifying the time windows (sequence lengths) and the values used for the tempering parameter. These details clarify the operational efficiency of our approach in dynamic scenarios.

---

> > ### Comment · Reviewer_RNwV · 2024-11-26
> >
> > Thank you for the answer. I still do not see the new section in the appendix, it looks like it is not added.

---

> ### Author Response · Authors · 2024-11-24
> **Response to The Reviewer's Comments on Weaknesses 2/3**
>
> > **Weakness 2:** *Some justification in picking the transformer-based encoder will be helpful. Is there a real benefit of using it, given the overhead and the large amount of data needed to train it? Why using it, if afterwards they have to be reduced to variational encoders with lower performance, and with tempering parameters?.*
>
>
> **Answer:** We appreciate the reviewer’s comment on the use of a transformer-based encoder and its implications. We have broken down the concern into specific points:
>
> 1. *Some justification in picking the transformer-based encoder will be helpful. Is there a real benefit of using it, given the overhead and the large amount of data needed to train it?*.
>     - The motivation behind using a transformer-based encoder arises Takens’s theorem, which suggests that using delayed coordinates can capture the dynamics more accurately. Empirically, we demonstrate that this method yields richer and better embeddings. As for the overhead of using a transformer-based encoder, we note that we use the same number of parameters for all of our models (about 45k parameter). In fact, our transformer is faster to train and infer than some baselines such as DKO [1] or the EMLP baselines due to parallelization capabilities. Transformers are known to scale well, meaning that given more data, bigger model size, and more compute, they usually give better results. However, we maintained a moderate dataset, and almost an equal model size to the other methods to ensure a fair comparison.
> 2. *Why using it, if afterwards they have to be reduced to variational encoders with lower performance*
>    - Here, we would like to make a distinction between the transformer encoder (we call it a *trajectory encoder*), and the transformer decoder (we call it the *action encoder*). The transformer encoder, which is the *trajectory encoder*, encodes the past states-action pairs, and is always of a non-variational type. Meanwhile, the *action encoder*, responsible for future action embeddings, can be variational or non-variational. The reason for this design choice is that the architecture is intended to be used in a sampling-based motion-planning frameowork. Essentialy, we sample a large number of actions and use the learned dynamic models to infer the states of the system in the future horizon. Then, using a cost function, we are able to select the best plan/path similar to [2]. It's worth emphasizing that a sampling-based planner samples huge number of actions (in the order of hundreds of thousands to over a million action samples in the case of autonomous driving). Using a non-variational transformer decoder or a feed-forward network, such as previous methods, would be computationally prohibitive because we would need to repeatedly query the neural network, which is not feasible for real-time applications. Instead, we only use the *action encoder (transformer decoder)* of a variational type to make the dynamically-feasible planning process real-time.
>
>    - Finally, although using a *variational action encoder* decreases performance, the transformer with the *variational action encoder* can be scaled to be much bigger than the normal transformer, and still be able to run in real-time. This yields better results (more accurate predictions) at the expense of larger model sizes. This is due to the fact that at each planning cycle, we will infer the transformer encoder *(trajectory encoder)*, only once to get the state embedding. Then, we sample the learned distribution to get different actions embeddings. Using both the state embeddings and different action embeddings, we can get different future states of the vehicle using the Koopman theory. In fact, a single future state inference is mainly two matrix multiplications. We also note that in our experiments, we kept the variational-based transformer of the same size of a non variational transformer for the sake of comparison. Using this algorithm, we are able to generate dynamically-feasible path/motion plans from a sampling-based planner. We added the pseudo code for our dynamically feasible path-planning in the appendix to clarify this procedure even further.
>
> 3. *and with tempering parameters?.*
>
> - As for the termpering parameters, they are used only in online adaptation to tackle distribution shifts. This is to make the matrix-normal distribution more capable of adapting to new online observations. Tempering only works on the probability distribution over the Koopman operator and not on the probability distribution over the embeddings (which results from the use of KL loss in the variational encoder). Tempering is used to decrease the confidence in the learned dynamics, which is the Koopman operator in our case. In other words, it decreases confidence in what the probabilistic Koopman model has learned before, to make online updates more effective.

---

> ### Author Response · Authors · 2024-11-24
> **Response to The Reviewer's Comments on Weaknesses 3/3**
>
> > **Weakness 3:** *It would be useful to see a comparison in the path planning comparison with an approach which is not based on Koopman-representation, such as hierarchical planning (A* and RRT) or policy optimization, or any other approach.*
>
> **Answer:** In our experiments, we integrated RRT* with the Koopman-representation to ensure the generated paths are dynamically feasible, meaning they can be physically followed by the vehicle. This aspect is crucial for articulated systems, such as a truck and trailer, and for challenging conditions like driving on ice. We have included the planning algorithm in the appendix in the revised version. Our future work will focus on dynamically-feasible planning and comparing our approach to reinforcement learning methods, such as policy optimization. We hope that this addresses the reviewer concern. Nevertheless, we are more than happy to discuss further details.
>
> ---
>
> References:
>
> [1] Shi, Haojie, and Max Q-H. Meng. "Deep Koopman operator with control for nonlinear systems." IEEE Robotics and Automation Letters 7.3 (2022): 7700-7707.
>
> [2] Pepy, Romain, Alain Lambert, and Hugues Mounier. "Path planning using a dynamic vehicle model." 2006 2nd International Conference on Information & Communication Technologies. Vol. 1. IEEE, 2006.

---

> ### Author Response · Authors · 2024-11-24
> **Response to The Reviewer's Questions**
>
> > **Question 1:** *Under the assumption that the noise vectors are i.i.d sampled from a multivariate Gaussian distribution, the learning of the Koopman operator using Bayesian LR model proceeds.*
> >    - *How important is this assumption? Is it valid realistically for autonomous driving?*
>
> **Answer:** The assumption that noise vectors are independent and identically distributed (i.i.d.) and sampled from a multivariate Gaussian distribution is a simplification commonly employed in Bayesian learning frameworks. While this assumption is generally reasonable, it may not fully capture the complexities of noise characteristics in real-world autonomous driving scenarios.
>
> 1. **Validity of the i.i.d. Gaussian Assumption**
>    - The i.i.d. Gaussian noise assumption is widely used because it facilitates efficient computation of Bayesian posteriors, which is essential for real-time decision-making in safety-critical applications..
>    - In autonomous driving, however, this assumption may not always hold true. Real-world noise characteristics often vary based on the sensor type and driving environment. For instance, some sensors may produce noise with temporal correlations, heteroscedasticity (non-constant variance), or multimodal distributions, especially under challenging conditions such as adverse weather or rapid dynamic changes.
>    - Despite its limitations, this assumption provides a practical trade-off between simplicity and performance, enabling effective uncertainty quantification in many scenarios.
>
> 2. **Potential Extensions to Non-i.i.d. Noise Models**
>    - To accommodate more realistic noise structures, the Bayesian method could be extended to handle non-i.i.d. noise using approaches such as Bayesian Model Averaging (BMA) or Mixture Models:
>      - **Bayesian Model Averaging (BMA)**: BMA accounts for model uncertainty by averaging over multiple models, allowing for better representation of diverse error distributions when noise structure is uncertain [1].
>      - **Mixture Models**: These combine multiple distributions to capture non-Gaussian errors, effectively modeling multimodal or skewed noise. For instance, mixtures of Gaussians can approximate complex patterns [2].
>    - However, incorporating non-i.i.d. noise significantly increases mathematical and computational complexity. As a result, these extensions are reserved for future research. We belive that our current method strikes a balance between accuracy and computational feasibility, but we aim to explore more complex models in future research.
>
> ---
>
> **References**
>
> - [1] Hoeting, J. A., Madigan, D., Raftery, A. E., & Volinsky, C. T. (1999). Bayesian Model Averaging: A Tutorial. *Statistical Science*, 14(4), 382-401.
>
> - [2] Frühwirth-Schnatter, S. (2006). *Finite Mixture and Markov-Switching Models*. New York: Springer-Verlag.

---

> ### Comment · Reviewer_RNwV · 2024-11-26
>
> W.2 This is a useful explanation. It would be good to see some of it either referenced, or in the manuscript.
>
> The rest is addressing my comments sufficiently. At this point, I am confident with my rating.

---

> ### Author Response · Authors · 2024-11-30
> **Key Revisions and Enhancements**
>
> We sincerely appreciate the reviewer’s thoughtful and constructive feedback. In response, we have made several updates and enhancements to strengthen our manuscript. Below is a summary of the key updates and improvements:
>
> ## Key Revisions Addressing the Reviewer's Concerns:
>
> - **Computational Assessment**:
>   A comprehensive computational evaluation has been conducted and added to **Appendix D: Additional Results** (pages 24–25).
>
> - **Transformer Encoder Justification**:
>   The rationale for employing the transformer encoder has been clarified in the **Methodology** section, located at the end of page 4. We have also highlighted the scalability feature of the variational action encoder in the **Experimental Results** section on page 9.
>
> - **Implementation Details**:
>   Detailed tables describing the implementation of our proposed method and baseline approaches are now included in **Appendix F** (pages 27–28).
>
> - **Motion Planning Details**:
>   Additional information on motion planning has been provided, including a pseudo-code algorithm for RRT-based motion planning utilizing our variational action sampling approach (**Appendix D**, pages 23–24).
>
> ---
>
> ## Additional Enhancements:
>
> - **Expanded Baseline Comparisons**:
>   We incorporated comparisons with additional baselines, including methods based on Koopman theory (e.g., Neural ODEs) and uncertainty quantification techniques such as Bayesian NNs and MC Dropouts. These updates are detailed in the **Experimental Setup** section and **Appendix D**.
>
> - **Loss Function Visualizations**:
>   Visual representations of loss functions across various baselines and environments have been added to the main experimental section and **Appendix D**.
>
> - **Real Vehicle Data Collection**:
>   **Appendix B** now includes a detailed explanation of the data acquisition and processing methods utilized in this study.
>
> - **Robustness Analysis**:
>   To demonstrate the robustness of our approach, we expanded the test dataset for truck dynamics to include challenging winter test data. This dataset features aggressive maneuvers absent from the training data and out-of-distribution scenarios (e.g., μ-split braking). These details are outlined in the **Experimental Setup** section (pages 7–10) and **Appendix B: Truck and Trailer Data Collected** (page 17).
>
> - **Ablation Studies**:
>   We conducted ablation studies to assess the impact of Bayesian learning and sequence length (history) on performance. Results for varying sequence lengths are included. This can be found in Appendix E.
>
> ---
>
> ## Supplementary Materials:
>
> - **Revised Manuscript**:
>   A fully revised manuscript with updates highlighted in blue.
>
> - **Open-Source Code**:
>   The accompanying open-source repository now includes detailed procedures for data collection and model training in simulated environments.
>
> We thank the reviewer for the constructive feedback. We believe these updates address the reviewer’s concerns and provide a more robust and comprehensive evaluation of our proposed method.
>
> We sincerely hope these revisions, along with our prior responses, will encourage a favorable re-evaluation of our manuscript. We remain fully open to any further feedback or questions the reviewer may have.

---

> > ### Comment · Reviewer_RNwV · 2024-12-02
> >
> > I appreciate the revision, especially the baselines and the computational times. I will adjust my rating.

---

> > > ### Author Response · Authors · 2024-12-02
> > >
> > > We deeply value the reviewer’s thoughtful and constructive feedback, along with their positive re-evaluation of our work. We remain fully committed to addressing any additional questions or insights the reviewer may have.

---

### Official Review · Reviewer_UD1y · 2024-11-03

**Soundness:** 3
**Presentation:** 3
**Contribution:** 3
**Rating:** 8
**Confidence:** 4

**Summary:**

In this paper, the authors propose a Koopman-based framework for robust motion planning for trucks. For modeling and prediction of nonlinear dynamics, an uncertainty-aware Koopman operator is introduced. The main idea is that a Bayesian regression model is utilized for the approximated operator, enabling uncertainty quantification based on its posterior distribution. Furthermore, distribution shifts (such as changing road conditions) are addressed by introducing a “changing variable” which detects potential shifts via a likelihood ratio test. Finally, the framework is integrated in a sampling-based motion planer.

The main contributions are i) a novel uncertainty-aware data-driven Koopman operator (using multi-steps) based on Bayesian methods, and ii) real-time adaptation via distribution shift detections.

**Strengths:**

- In general, the paper is well written, the approaches are well motivated, and the presentation of the methods is clear.

- Based on my knowledge, the modeling of the approximated Koopman operator via Bayesian methods, resulting in a Wishart distribution for the posterior is novel and original. Furthermore, be combining the Koopman operator theory with transformers and the variational autoencoder for motion planning seems to be a smart and beneficial way to address the problem of robust motion planning.

- The simulations results indicate that the proposed method can outperform other Koopman based methods and a standard MLP approach. In this way, it seems to be a significant improvement for this scenario based on this dataset.

**Weaknesses:**

1. The authors provide a detailed overview about the related state of the art. Reading this section, it seems that man of the “open challenges” that this paper addresses are already solved in some way in the existing literature. I assume that the authors do address existing gaps, but they missed to cleary point out these gaps. Me recommendation: Add some sentences in section 2 explaining the remaining research gaps.

2. 177: “Koopman […] under the assumption that the controls do not evolving dynamically” I’m not sure if I understand this statement correctly. My assumption is that “dynamically” refers to an input that depends on the state, i.e., a feedback controller. Even thought that might be a valid assumption, the title of the paper indicates the framework is for “autonomous trucks” where I assume we do have the feedback loop.

3. Eq (4): From a data-driven perspective with a noise data set, etc, the extension to a multi-step input and output seems to make sense. However, based on the original Koopman theory, the extension seems to be unnecessary.

4. While reading the paper, I stumble across Eq. (5). From my understanding, we do a non-linear mapping from the original non-linear dynamics to a high-dimensional but linear space. However, in eq (4), the authors propose to do a linear mapping form the lifted space to the original space. Maybe it is a misunderstanding, but that makes no sense to me. Furthermore, it is a general challenge to design the mapping that the inverse also exists. I do not see any evidence here that the inverse mapping might exist.

Minor:
- Citation style is often not correct (citep instead of citet, or the other way around)
-460: “Finally, While” -> while

**Questions:**

Weakness 2: Is this assumption still valid? What is the general problem with a dynamical input? Do you have an idea how to deal with it?

Weakness 3: Is there a theoretical justification for the multistep approach? Does it improve performance even for noise-free datasets? Could you provide more evidence on the benefit of this extension?

Weakness 4: Why is the inverse mapping model considered a linear function? Can you elaborate on the fact the inverse might even not exist?

---

> ### Author Response · Authors · 2024-11-24
> **Response to The Reviewer's Comments on Weaknesses 1/2**
>
> We thank the reviewer for the thoughtful and constructive feedback. We address each concern below.
>
> > **Weakness 1:** *The authors provide a detailed overview about the related state of the art. Reading this section, it seems that man of the “open challenges” that this paper addresses are already solved in some way in the existing literature. I assume that the authors do address existing gaps, but they must describe to clearly point out these gaps. Me recommendation: Add some sentences in section 2 explaining the remaining research gaps.*
>
>
> **Answer:** We sincerely thank the reviewer for the thoughtful suggestion. Our paper addresses several key areas:
>
> 1. **Long-Horizon Predictions:** Unlike previous methods such as [1] and [2], which are limited to short-horizon models (e.g., 16 time steps), our approach is designed for long-horizon predictions, crucial for our application. This necessitates more accurate dynamic modeling and richer embeddings, which we achieve by embedding entire trajectories rather than individual timesteps.
>
> 2. **Uncertainty Estimation:** Existing methods [1], [4] typically model uncertainty by placing a probability distribution over the embedded (lifted) space, which does not capture uncertainties in the system dynamics. In contrast, our approach places a probability distribution directly over the Koopman operator, providing a more robust and comprehensive framework for modeling both aleatoric and epistemic uncertainty.
>
> 3. **Distribution Shifts:** Prior works such as [3] has focused on distribution shifts primarily in time-series data without control inputs. Our method, however, addresses distribution shifts in dynamical systems that include control inputs, making it more applicable to real-world scenarios.
>
> 4. **Motion Planning:** While most existing methods are used for learning world models or developing control-suitable models using Koopman theory, our approach uniquely applies this framework to motion planning for autonomous vehicles.
>
> We have clarified these research gaps in the introduction of our paper.
>
> ---
>
> > **Weakness 2:** 177: “Koopman [...] under the assumption that the controls do not evolving dynamically” I’m not sure if I understand this statement correctly. My assumption is that “dynamically” refers to an input that depends on the state, i.e., a feedback controller. Even though that might be a valid assumption, the title of the paper indicates the framework is for “autonomous trucks” where I assume we do have the feedback loop.
>
> **Answer:** This is an excellent point. In fact, an autonomous truck operates within a closed feedback loop, so the controls do evolve dynamically. We have revised the manuscript to reflect the dynamic evolution of the system.
>
> ---
>
> > **Weakness 3:** Eq (4): From a data-driven perspective with a noise data set, etc, the extension to a multi-step input and output seems to make sense. However, based on the original Koopman theory, the extension seems to be unnecessary.
>
> **Answer:** We agree with the reviewer that, from a Koopman theory standpoint, this extension is not essential for a fully-observed system. However, we have indeed observed notable improvements from this extension in both noisy and noise-free environments, as the reviewer also highlighted. To further validate this, we plan to perform ablation studies by disabling Bayesian learning and testing various sequence lengths, including a sequence length of one (equivalent to a single-time step embedding). Detailed results will be provided in the appendix of the next revised version. If the reviewer believes an alternative experimental setup would be more appropriate, we would be happy to accommodate it. We thank the reviewer for the insightful comments.
>
> ---

---

> ### Author Response · Authors · 2024-11-24
> **Response to The Reviewer's Comments on Weaknesses 2/2**
>
> > **Weakness 4:** While reading the paper, I stumble across Eq. (5). From my understanding, we do a non-linear mapping from the original non-linear dynamics to a high-dimensional but linear space. However, in eq (4), the authors propose to do a linear mapping from the lifted space to the original. Maybe it is a misunderstanding, but that makes no sense to me. Furthermore, it is a general challenge to design the mapping that the inverse also exists. I do not see any evidence that the inverse mapping might exist.
>
>
> **Answer:** We thank the reviewer for bringing up this important point. We will break down the concern into several parts and address each one separately.
>
> > *While reading the paper, I stumble across Eq. (5). From my understanding, we do a non-linear mapping from the original non-linear dynamics to a high-dimensional but linear space. However, in eq (4), the authors propose to do a linear mapping from the lifted space to the original. Maybe it is a misunderstanding, but that makes no sense to me.*
>
> We clarify that in Eq. (4), our method performs a nonlinear mapping from the original state $ [x, u]^\top $ to a high-dimensional embedding space $\tilde{z}$. This transformation lifts an entire trajectory at once to create richer and more informative embeddings, which is essential for accurately capturing system dynamics. In contrast, Eq. (5) describes the reverse process, where we map from the lifted space back to the original state space. Importantly, we assume that the lifted space can be decomposed into two components: the state embedding $\tilde{x}$ and the action embedding $\tilde{u}$, such that $\tilde{z} = [\tilde{x}; \tilde{u}]$. When we project from the lifted space to the original, we specifically focus on the state embedding subspace. Mathematically, this is given by Eq. (5).
>
> > *Furthermore, it is a general challenge to design the mapping that the inverse also exists. I do not see any evidence that the inverse mapping might exist.*
>
> The reviewer raises a valid concern regarding the challenge of ensuring the existence of an inverse mapping. While it is indeed true that an exact inverse may not always be attainable, our approach is specifically designed to optimize (learn) the mapping function so that the generated embeddings allow for a linear inverse mapping that is either feasible or approximately accurate. Although this approximation introduces some error, it is a necessary trade-off to ensure the computational efficiency required for real-time performance.
>
> ---
>
> **Minor Weakness:** Citation style is often not correct (citep instead of citet, or the other way around) -460: “Finally, While” -> while
>
> **Answer** We thank the reviewer for pointing this out. It has been fixed in the manuscript.
>
>
> ---
>
> References:
>
> - [1] Han, Minghao, Jacob Euler-Rolle, and Robert K. Katzschmann. "DeSKO: Stability-assured robust control with a deep stochastic Koopman operator." International Conference on Learning Representations. 2021.
> - [2] Shi, Haojie, and Max Q-H. Meng. "Deep Koopman operator with control for nonlinear systems." IEEE Robotics and Automation Letters 7.3 (2022): 7700-7707.
> - [3] Wang, Rui, et al. "Koopman neural operator forecaster for time-series with temporal distributional shifts." The Eleventh International Conference on Learning Representations. 2023.
> - [4] Frion, Anthony, et al. "Koopman Ensembles for Probabilistic Time Series Forecasting." 2024 32nd European Signal Processing Conference (EUSIPCO). IEEE, 2024.

---

> ### Author Response · Authors · 2024-11-24
> **Response to The Reviewer's Questions**
>
> We appreciate the insightful questions and address them as follows:
>
> > **Question 1** *Weakness 2: Is this assumption still valid? What is the general problem with a dynamical input? Do you have an idea how to deal with it?*
>
> **Answer:** We address the reviewer's question point by point below:
>
> > Weakness 2: Is this assumption still valid?
>
> No, we believe it should instead be dynamically evolving. We have updated the text to reflect this change.
>
> > What is the general problem with a dynamical input? Do you have an idea how to deal with it?
>
> We do not see a general issue with the use of a dynamical input and have updated the manuscript to reflect this perspective.
>
> ---
>
> > **Question 2** Weakness 3: Is there a theoretical justification for the multistep approach? Does it improve performance even for noise-free datasets? Could you provide more evidence on the benefit of this extension?
>
> **Answer:** The only theoretical justification we can provide is based on Taken's theorem. Empirically, we have observed that this approach enhances performance on both noisy and noise-free datasets. To further validate this, we plan to conduct ablation studies by disabling Bayesian learning and experimenting with different sequence lengths, including a sequence length of one (equivalent to a single-time step embedding). We will include these detailed results in the appendix of the revised version. Additionally, we are open to conducting any other experiments the reviewer may suggest.
>
> ---
>
> > **Question 3** Weakness 4: Why is the inverse mapping model considered a linear function? Can you elaborate on the fact the inverse might even not exist?
>
> **Answer:** The choice of using a linear function for the inverse mapping is primarily driven by computational efficiency. Formally, the inverse mapping should also be learned using another neural network. Although it’s true that an exact a linear inverse mapping may not always be achievable, our approach is designed to learn a mapping function in a way that supports a feasible or approximately accurate linear inverse. This approximation does introduce some error, but it is a necessary trade-off to meet the real-time performance requirements.
>
>
> ---
>
> We hope that this answers the reviewers insightful questions.

---

> > ### Comment · Reviewer_UD1y · 2024-11-27
> >
> > Thank you very much for the detailed feedback. I truly appreciate that the authors addressed all my concerns. I’m still surprised that a linear approximation for the inverse mapping works but, obviously, it does at least in this case.
> >
> > In conclusion, I recommend to accept the paper for presentation at ICLR.
> >
> > Looking forward to future work of the authors.

---

> > > ### Author Response · Authors · 2024-11-30
> > > **Official Comment By The Authors**
> > >
> > > We sincerely appreciate the reviewer’s thoughtful and constructive feedback, as well as their positive assessment of our work. We remain fully open to addressing any further questions or insights the reviewer may have.

---

### Official Review · Reviewer_K4gS · 2024-11-04

**Soundness:** 2
**Presentation:** 2
**Contribution:** 2
**Rating:** 3
**Confidence:** 2

**Summary:**

This paper deals with Koopman theory, where physical dynamics are modelled with data. To account for adapting dynamics, this paper proposes a Bayesian formulation. For example a model of a truck on desert or in snow results in different dynamics due to the distribution shifts, and the paper tries to incorporate uncertainty measures. The method is tested on a real data of a truck, validating the proposed framework.

**Strengths:**

Indeed, Koopman theory is one of the widely examined topic in robotics, and there, one needs to account for varying dynamics and incorporate uncertainty.

**Weaknesses:**

However, the paper needs to distinguish better between existing works. There has been active learning paradigms for Koopman theory, and how is this work better or differs? Those aspects should be taken into account for the list of contributions.
When compared to papers in robotic conferences, I think the paper fall short in terms of experimental evaluation, e.g., the paper uses a relatively small data set for the truck dynamics, while Koopman theory is more for learning complicated dynamics like soft robotic manipulators. Moreover, real world experiment should be there to indicate that the proposed method works in practice.
The paper’s topic might not also perfectly fit ICLR but rather IROS and ICRA.

**Questions:**

How valid is the prior being used here? Would there also be ways to incorporate the underlying physics more than isotropic Gaussians?

---

> ### Author Response · Authors · 2024-11-24
> **Response to The Reviewer's Comments on Weaknesses 1/2**
>
> We appreciate the reviewer’s feedback and the opportunity to clarify the unique contributions and merits of our work. We address each part of the weaknesses highlighted as follows:
>
> > **Weakness 1:** *However, the paper needs to distinguish better between existing works. There has been active learning paradigms for Koopman theory, and how is this work better or differs? Those aspects should be taken into account for the list of contributions.*
>
> **Answer:** We acknowledge the active research on modeling physical systems using Koopman theory. However, our approach addresses gaps in current methods as follows:
> - **Uncertainty Quantification:** Prior methods predominantly focus on modeling uncertainty in the embedded latent space by placing a probability distribution over the observables or embedding vectors [1, 2, 3]. Our method places a probability distribution over the Koopman operator itself. This formulation enables us to reason systematically about both aleatoric (data) and epistemic (model) uncertainties, which is crucial for robust real-world applications.
> - **Handling Distribution Shifts:** Existing Koopman-based methods do not adequately address distributional shifts, especially for dynamic systems with control inputs [4]. By leveraging Bayesian learning, our approach adapts to changes in system dynamics in real time, ensuring robustness in scenarios with significant temporal and environmental variations.
> - **Novel Application in Motion Planning:** While previous works have concentrated on control or RL applications [5, 6], our method extends Koopman theory to generate dynamically feasible path plans. This application is critical for autonomous systems where both motion feasibility and real-time adaptability is essential.
>
> We have incorporated a dedicated section in the revised manuscript to clearly highlight our method’s contributions compared to the Koopman theory literature.
>
> ---
>
> > **Weakness 2:** *When compared to papers in robotic conferences, I think the paper falls short in terms of experimental evaluation, e.g., the paper uses a relatively small data set for the truck dynamics, while Koopman theory is more for learning complicated dynamics like soft robotic manipulators.*
>
> **Answer:** We agree with the reviewer that extensive evaluation is cruical. Our experimental evaluation leverages six distinct datasets: a real-world truck dynamics dataset and five additional simulated environments. The real-world dataset comprises over 10 hours of processed driving data, collected under diverse and challenging conditions and gathered specifically for this work. We further augmented the test dataset with data collected during a winter test performed on a specialized test tracks in northern Sweden, specifically addressing low-traction scenarios like icy and snowy roads. This ensures rigorous testing under a range of real-world conditions. Additionally, the five simulated environments, constructed using the MuJoCo physics engine, provide further validation of our method’s robustness. For each simulation, we run it with both process noise and observation noise to comprehensively test our approach. We have clarified this in the paper. Moreover, in the revised version, more visualizations will be added as well as experiments for distribution shifts. Finally, we note that the Koopman theory addresses general dynamical systems and not only soft robotics. For example, [3], [5], [6], [7] use the Koopman theory for modeling simulated physical systems in Mujoco, similar to our simulated environments. This is due to the present dynamical complexity in such systems. We hope that the reviewer is satisfied with these updates. However, we welcome any further comments and requests.
>
> ---
>
> > **Weakness 3:** *Moreover, real-world experiment should be there to indicate that the proposed method works in practice.*
>
> **Answer:** We agree that real-world vehicle experiments are essential. We note that our paper includes evaluations based on extensive real-world experiments, as described above.

---

> ### Author Response · Authors · 2024-11-24
> **Response to The Reviewer's Comments on Weaknesses 2/2**
>
> ---
>
> > **Weakness 4:** *The paper’s topic might not also perfectly fit ICLR but rather IROS and ICRA.*
>
> **Answer:** We thank the reviewer for bringing up this important point. We believe our work fits well within ICLR’s scope, as it focuses on learning Koopman embeddings for real-time motion planning—essentially, learning representations of dynamical systems with practical applications in motion planning. ICLR has published numerous papers on using the Koopman theory to model physical systems as well as works related to representation learning for planning and reinforcement learning (RL). Examples of these works published in ICLR are: [1], [7], [8].
>
> ---
>
> Refernces:
>
> - [1] Han, Minghao, Jacob Euler-Rolle, and Robert K. Katzschmann. "DeSKO: Stability-assured robust control with a deep stochastic Koopman operator." International Conference on Learning Representations. 2021.
> - [2] Morton, Jeremy, Freddie D. Witherden, and Mykel J. Kochenderfer. "Deep variational koopman models: Inferring koopman observations for uncertainty-aware dynamics modeling and control." arXiv preprint arXiv:1902.09742 (2019).
> - [3] Frion, Anthony, et al. "Koopman Ensembles for Probabilistic Time Series Forecasting." 2024 32nd European Signal Processing Conference (EUSIPCO). IEEE, 2024.
> - [4] Wang, Rui, et al. "Koopman neural operator forecaster for time-series with temporal distributional shifts." The Eleventh International Conference on Learning Representations. 2023.
> - [5] Shi, Haojie, and Max Q-H. Meng. "Deep Koopman operator with control for nonlinear systems." IEEE Robotics and Automation Letters 7.3 (2022): 7700-7707.
> - [6] Mondal, Arnab Kumar, et al. "Efficient Dynamics Modeling in Interactive Environments with Koopman Theory." arXiv preprint arXiv:2306.11941 (2023).
> - [7] Fathi, Mahan, et al. "Course Correcting Koopman Representations." arXiv preprint arXiv:2310.15386 (2023).
> - [8] Mohammadi, Hadi Beik, et al. "Neural Contractive Dynamical Systems." The Twelfth International Conference on Learning Representations.

---

> ### Author Response · Authors · 2024-11-24
> **Response to The Reviewer's Questions**
>
> We thank the reviewer for raising these questions. We address them as follows:
>
> > **Question 1:** *How valid is the prior being used here?*
>
>
> **Answer:** The use of the Matrix Normal Inverse Wishart (MNIW) distribution as a prior for the Koopman operator and the associated covariance matrix is grounded in several principles that make it particularly well-suited for our problem. Below, we discuss the mathematical justification and relevance of our chosen prior as well as the benefits of using it.
>
> 1. **Uncertainty Quantification**:
>    - The MNIW distribution captures the joint variability in both the Koopman operator and the covariance matrix. This is important for robust modeling of the underlying dynamics, as it provides a comprehensive representation of uncertainty in both the system's dynamics and the associated noise structure.
>    - Our choice of prior allows us to simultaneously model both aleatoric uncertainty (uncertainty inherent in the stochastic system) and epistemic uncertainty (uncertainty due to limited data or model knowledge). This dual modeling capability is crucial for systems with dynamic and evolving behavior.
>
>
> 2. **Central Limit Theorem**:
>    - Many physical and stochastic systems can be approximated well by Gaussian models due to the Central Limit Theorem. In our case, modeling the Koopman operator as a random matrix with a Gaussian structure (through the Matrix Normal component) aligns with the idea that the underlying dynamics can be approximated as a linear system with Gaussian noise.
>    - The Inverse Wishart component is a standard choice for modeling uncertainty in covariance matrices, given that it is the conjugate prior for multivariate Gaussian distributions and is widely used in Bayesian hierarchical models. The combination of the Matrix Normal and Inverse Wishart distributions provides a mathematically sound and interpretable framework for our Bayesian modeling approach.
>
> 3. **Conjugate Prior**:
>    - The MNIW prior is a conjugate prior for matrix-variate Gaussian likelihood models. This conjugacy simplifies the mathematical formulation and allows for efficient updates of the distribution parameters using Bayesian inference. Specifically, in our case, the conjugate nature of the MNIW distribution facilitates efficient learning and updating of the Koopman operator under the probabilistic framework, while maintaining analytical tractability.
>    - The use of conjugate priors like the MNIW is a well-established practice in Bayesian statistics, as it allows for closed-form posterior updates, which are crucial for computational efficiency and theoretical consistency, especially in systems where real-time processing is needed.
>
> 4. **Distribution Shifts**:
>    - One of the key advantages of the MNIW prior is its flexibility in addressing distribution shifts. When a system's dynamics evolve over time or exhibit non-stationary behavior, the ability to systematically update the prior distribution with new data is critical. The MNIW distribution provides a well-defined way to incorporate observed data and adapt the model parameters, ensuring that our approach remains robust and responsive to changes.
>    - This adaptability is particularly important for real-world applications, where distributional assumptions may not remain static, and it underscores the practical significance of our approach.
>
> ---
>
> > **Question 2:** *Would there also be ways to incorporate the underlying physics more than isotropic Gaussians?*
>
> **Answer:** Thank you for your insightful question. Incorporating the underlying physics more rigorously, beyond the use of isotropic Gaussian models, represents a potential extension of our work. One promising approach is to integrate physics-informed priors into the Koopman operator framework, similar to [1], which utilizes automatic differentiation to enforce physical laws through soft penalty constraints during training. This method embeds known dynamics directly into the model, reducing the need for extensive datasets and improving both data efficiency and predictive performance.
>
> In future works, we are considering approaches where the underlying physics can serve as the mean of a matrix-normal distribution over the Koopman operator. However, this remains an open area of research, and we plan to further explore and refine these ideas to better capture complex, nonlinear dynamics and extend the model's predictive performance.
>
> ---
>
> References:
>
> - [1] Liu, Yuying, et al. "Physics-informed koopman network." arXiv preprint arXiv:2211.09419 (2022).

---

> ### Author Response · Authors · 2024-11-30
> **Key Revisions and Enhancements**
>
> We sincerely thank the reviewer for their thoughtful and constructive feedback. In response, we have implemented the following updates and improvements:
>
> ## Key Revisions Addressing the Reviewer's Concerns:
>
> - **Differentiation from Existing Work**:
>   We have clearly stated how our method differs from existing approaches in the literature. These clarifications are detailed in the **Related Work** section on page 3.
>
> - **Enhanced Evaluation**:
>   - Comparisons with additional baselines, including Koopman theory methods (e.g., Neural ODEs) and uncertainty quantification approaches (e.g., Bayesian NN and MC Dropouts), have been incorporated. These results are presented in the **Experimental Setup** section and **Appendix D (Additional Results)**.
>   - Visualizations of loss functions across various baselines and environments have been added to both the main experimental section and **Appendix D**.
>   - A runtime analysis comparing our model to other baselines is now included in **Appendix D**.
>
> - **Robustness in Poor and Out-of-Distribution Scenarios**:
>   To showcase the robustness of our approach, we enhanced the test dataset for truck dynamics by limiting it to winter test data. This dataset includes aggressive maneuvers absent in the training data and out-of-distribution scenarios, such as braking in a μ-split scenario. These updates are discussed in the **Experimental Setup** section (pages 7–10) and **Appendix B (Truck and Trailer Data Collected)** on page 17.
>
> ---
>
> ## Additional Enhancements:
>
> - **Real Vehicle Data Collection**:
>   **Appendix B** now includes an in-depth explanation of the data acquisition and processing methods used in the study.
>
> - **Motion Planning Details**:
>   Additional implementation details related to motion planning are provided in **Appendix D**.
>
> - **Implementation Details**:
>   **Appendix F** now includes a comprehensive description of the implementation specifics for our method and the baseline models.
>
> - **Ablation Studies**:
>   Ablation studies were conducted to evaluate the benefits of Bayesian learning and to examine how sequence length (history) influences performance. These results are presented for varying sequence lengths. This can be found in **Appendix E**.
>
> ---
>
> ## Supplementary Materials:
>
> - A revised manuscript with all updates clearly highlighted in blue.
> - Open-source code covering the data collection process and model training procedures for the simulated environments.
>
> We deeply appreciate the reviewer’s feedback. We hope these updates and newly incorporated materials sufficiently address the reviewer’s concerns and provide a clearer demonstration of the potential and robustness of our proposed method.
>
> We hope these revisions will encourage the reviewer to re-evaluate our work positively, and we remain open to any further feedback or questions that could help us further refine our contribution.

---

### Official Review · Reviewer_QDfV · 2024-11-04

**Soundness:** 3
**Presentation:** 3
**Contribution:** 2
**Rating:** 5
**Confidence:** 3

**Summary:**

The authors propose a Bayesian framework for learning a Koopman operator-based predictive model. The model takes states and actions as inputs, allowing it to be used for planning. The authors use a transformer architecture to map the state to an embedding vector. They then use a Bayesian approach to formulate the distribution of the Koopman operator and that of the mapping from latent to state space. The posterior can be computed analytically given the data. To be able to sample efficiently during planning, the authors use a variational auto-encoder, which allows the action to be sampled in the latent space directly, as this corresponds to sampling from a Gaussian.

**Strengths:**

The paper is well written and easy to read. The problem is well motivated and the literature overview is adequate.

**Weaknesses:**

The proposed method has no theoretical guarantees. It would be interesting to at least have a discussion on what to expect without any formal result.

The algorithmic contribution is not very significant, as it consists of building blocks taken from existing methods.

**Questions:**

Line 145/146 has a typo.

Line 331: why does reducing the prior variance incur a broader posterior? Intuitively, the opposite is true. Do the authors mean increasing instead of reducing?

Is the method comparable to an approach that uses a nonlinear function to propagate the dynamics in the latent space, e.g., "Dream to Control: Learning Behaviors by Latent Imagination"?

How accurate  are model predictions using out-of-distribution data? An assessment with a corresponding distinction would be useful.

Can the proposed approach be used for reinforcement learning?

How does the approach compare to predictive approaches other than Koopman? It would be interesting to see how a one-step predictive method using a Bayesian neural network or Gaussian process performs.

In the appendix, the authors state that the data is collected using a TD3 agent. I feel that this is relevant and  should be mentioned in the main body of text.

Though it only uses a Gaussian process instead of a transformer, the paper "Gaussian Process-Based Representation Learning via Timeseries Symmetries" also provides a measure of model uncertainty. How does this compare to the proposed approach?

How well does the approach perform if the collected data is poor? How does the model perform out of distribution?

How well does the method scale? Eq. (11)-(13) indicate that the Gram matrix of the data needs to be inverted to compute the posterior, which scales cubically with the amount of data.

---

> ### Author Response · Authors · 2024-11-24
> **Response to The Reviewer's Comments on Weaknesses**
>
> We thank the reviewer for bringing up these points. We address the concerns raised as follows:
>
> > **Weakness 1:** *The proposed method has no theoretical guarantees. It would be interesting to at least have a discussion on what to expect without any formal result.*
>
> **Answer:** Our framework follows a curriculum learning paradigm of two stages: (1) learning embeddings with transformers, and (2) Bayesian learning to model the stochastic Koopman operator and account for uncertainty. In the first stage, the use of transformers to learn embeddings follows established machine learning practices. The quality of these embeddings is assessed using a loss function, as is typical in deep learning models, and guarantees here align with those generally expected in machine learning, where a minimized loss value indicates that the model has captured the underlying dynamics effectively. Nevertheless, imperfections in the embeddings are dealt with in the second stage of our framework (Bayesian learning). In the second stage, we apply Bayesian learning using both the learned embeddings and the ground truth states of the system. This is crucial because even if the learned embeddings are not perfect, the Bayesian framework is designed to handle these imperfections. By placing probability distributions over the Koopman operator, Bayesian learning accounts for the uncertainty introduced by imperfect embeddings (the first stage), leading to more robust and uncertainty-aware predictions. We provide formal derivations in the appendix for both the posterior distribution (the model parameters) and the posterior predictive distribution (the predictions of the system states given the uncertainty in model parameters and the imperfect embeddings), ensuring our approach is well-grounded. Finally, we apply our method to the problem of generating dynamically feasible path plans by sampling the embedded space directly for computational efficiency. Here, our path-planning approach, akin to other sampling-based planners, offers asymptotic guarantees—meaning optimality is approached as the number of samples increases. We will clarify this in the revised version.
>
> ---
>
> > **Weakness 2:** *The algorithmic contribution is not very significant, as it consists of building blocks taken from existing methods.*
>
>
> **Answer:** We appreciate the reviewer’s thoughtful feedback and would like to clarify the contributions of our work. Our approach introduces a novel way to learn Koopman operators. Specifically, we model the Koopman operator with a joint probability distribution—using a Matrix Normal Inverse Wishart distribution—over the Koopman matrix and the associated covariance matrix. We believe that this is a new way for learning Koopman operators and system dynamics, and leads to new algorithms. Existing approaches, for example [1, 2, 3], model the uncertainty by placing a probability distribution over the embedded vector. Our method allows for rigorous uncertainty quantification (both aleatoric and epistemic uncertainties) as well as addressing distribution shifts in a systematic manner. We show that this approach is suitable for our application, but we also believe that it is applicable for many other real-world applications that contain sources of uncertainty or time changing dynamics. Additionally, our approach to embedding full trajectories rather than individual state-action pairs provides richer context and has not been explored before in this setting, offering improved modeling of the dynamics. Finally, we leverage the variational autoencoders (VAEs) to allow dynamically-feasible real-time motion planning. We hope that this addresses the reviewer's concern. We have updated the paper and clarified our key contributions. We would be glad to address any additional questions or concerns the reviewer may have.
>
> ---
>
> References:
>
> 1. Han, Minghao, Jacob Euler-Rolle, and Robert K. Katzschmann. "DeSKO: Stability-assured robust control with a deep stochastic Koopman operator." International Conference on Learning Representations. 2021.
> 2. Morton, Jeremy, Freddie D. Witherden, and Mykel J. Kochenderfer. "Deep variational koopman models: Inferring koopman observations for uncertainty-aware dynamics modeling and control." arXiv preprint arXiv:1902.09742 (2019).
> 3. Frion, Anthony, et al. "Koopman Ensembles for Probabilistic Time Series Forecasting." 2024 32nd European Signal Processing Conference (EUSIPCO). IEEE, 2024.

---

> ### Author Response · Authors · 2024-11-24
> **Response to The Reviewer's Questions 1/2**
>
> We thank the reviewer for raising these important questions. Here, we try to answer these questions:
>
> > 1. **Question:** *Line 145/146 has a typo.*
>
> **Answer:** We thank the reviewer for pointing out this typo. We have fixed it in the revised version.
>
> > 2. **Question:** *Line 331: why does reducing the prior variance incur a broader posterior? Intuitively, the opposite is true. Do the authors mean increasing instead of reducing?*
>
> **Answer:** Indeed, increasing the variance results in a broader distribution and not the opposite. We thank the reviewer very much for pointing this out.
>
> > 3. **Question:** *Is the method comparable to an approach that uses a nonlinear function to propagate the dynamics in the latent space, e.g., "Dream to Control: Learning Behaviors by Latent Imagination"?*
>
> **Answer:** Yes, we think that these methods can be related in the context of reinforcement learning. More specifically, our method can be used to learn “world models” in model-based reinforcement learning.
>
> > 4. **Question:** *How accurate are model predictions using out-of-distribution data? An assessment with a corresponding distinction would be useful.*
>
> **Answer:** We agree with the reviewer that out-of-distribution experiments would further give insights about our method. Out-of-distribution experiments will be added in the appendix in the revised version.
>
> > 5. **Question:** *Can the proposed approach be used for reinforcement learning?*
>
> **Answer:** Our approach can indeed be used for reinforcement learning in multiple ways. One way is to leverage uncertainty estimates in decision-making, the model could help guide exploration in RL settings. We have elaborated on this potential extension in the paper and discuss how the Koopman framework could be beneficial in the context of RL tasks, particularly in environments where dynamics are stochastic and require adaptive planning. Moreover, our model can be used to learn “world models” in model-based RL. Similar works already exist in the literature. Note that, both our method and [1] (PETS) leverage uncertainty-aware dynamics models; however, while PETS uses an ensemble approach combined with trajectory sampling for model-based RL, our approach uses a Bayesian Koopman framework focused on quantifying uncertainty and sampling from a the embedding space for real-time planning tasks.
>
> > 6. **Question:** *How does the approach compare to predictive approaches other than Koopman? It would be interesting to see how a one-step predictive method using a Bayesian neural network or Gaussian process performs.*
>
> **Answer:** We will include a comparison with Bayesian neural networks, Monte Carlo dropout, and Neural ODEs in the revised version. It is important to note that models trained solely for one-step predictions often perform poorly when evaluated over multiple steps. For instance, in our experiments, we use an ensemble of neural networks as a baseline, where a multi-step training objective is essential to achieve good long-term prediction accuracy. Single-step training objectives typically degrade performance over extended horizons, a trend that has been observed consistently in our experience and also in prior research. Therefore, both our method and the baselines adopt multi-step training strategies, as supported by evidence from the literature [2, 3, 4]. If the reviewer wants to see single-step predictions performance to provide more insights to the reader we will be happy to include that.

---

> ### Author Response · Authors · 2024-11-24
> **Response to The Reviewer's Questions 2/2**
>
> > 7. **Question:** *In the appendix, the authors state that the data is collected using a TD3 agent. I feel that this is relevant and should be mentioned in the main body of text.*
>
> **Answer:** We thank the reviewer for bringing up this important point. We will mention that in the main body of the paper in the revised version.
>
> ---
>
> > 8. **Question:** *Though it only uses a Gaussian process instead of a transformer, the paper "Gaussian Process-Based Representation Learning via Timeseries Symmetries" also provides a measure of model uncertainty. How does this compare to the proposed approach?*
>
> **Answer:** Both methods leverage Koopman theory to model dynamical systems, but there are key differences. The Gaussian Process-based approach focuses on learning latent linear time-invariant (LTI) dynamics using Gaussian Processes, which allows for closed-form uncertainty quantification and efficient multi-step prediction. In contrast, our method uses a Bayesian framework with transformers to represent complex, high-dimensional dynamics and quantify uncertainty, specifically designed for adaptive planning. Additionally, our approach prioritizes real-time adaptability and robustness in uncertain environments, while their method emphasizes symmetry-based regularization and tractable continuous-time posteriors.
>
> ---
>
> > 9. **Question:** *How well does the approach perform if the collected data is poor? How does the model perform out of distribution?*
>
> **Answer:** If the data quality is poor, the uncertainty estimates from our Bayesian approach become critical. In our experiments, we tested the model in various simulated environments under both process and observation noise to assess its robustness. We will add open-loop visualizations for various environments to show the effect of uncertainty estimation in the revised version.
>
> ---
>
> > 10. **Question:** *How well does the method scale? Eq. (11)-(13) indicate that the Gram matrix of the data needs to be inverted to compute the posterior, which scales cubically with the amount of data.*
>
> **Answer:** We thank the reviewer for bringing up this important point. Here, the inverse operation is performed on the $S_{zz}$ matrix. This matrix has dimension of $d \times d$, where $d$ is the dimension of the embedding and not the number of data points. Typically, the embedding dimension is relatively small ($d < 1024$), and in our experiments, we use $d= 32$. Additionally, this computational overhead occurs only during training, making it manageable and not a bottleneck in practice.
>
> ---
>
> **References:**
> 1. Chua, Kurtland, et al. "Deep reinforcement learning in a handful of trials using probabilistic dynamics models." Advances in neural information processing systems 31 (2018).
> 2. Mondal, Arnab Kumar, et al. "Efficient Dynamics Modeling in Interactive Environments with Koopman Theory." arXiv preprint arXiv:2306.11941 (2023).
> 3. Fathi, Mahan, et al. "Course Correcting Koopman Representations." arXiv preprint arXiv:2310.15386 (2023).
> 4. Han, Minghao, Jacob Euler-Rolle, and Robert K. Katzschmann. "DeSKO: Stability-assured robust control with a deep stochastic Koopman operator." International Conference on Learning Representations. 2021.

---

> ### Author Response · Authors · 2024-11-30
> **Key Revisions and Enhancements**
>
> We sincerely thank the reviewer for their valuable feedback, which has been very beneficial in improving our manuscript. In response, we have made the following updates and additions:
>
> ## Key Updates and Clarifications:
>
> - **Contributions and Novelty**:
>   We have clearly stated the unique contributions of our work, explicitly distinguishing it from existing methods in the literature. These clarifications are detailed in the **Background & Related Work** section on page 3.
>
> - **Theoretical and Algorithmic Contributions**:
>   The theoretical aspects and the algorithmic innovations of our method have been further clarified and emphasized. These updates are located in the **Introduction** section on page 2.
>
> - **Data Collection Details**:
>   We now explicitly mention the use of the TD3 agent for collecting the dataset in simulated environments  in the main body of text. This is described at the end of page 8 in the **Experimental Results** section.
>
> - **Potential for RL Integration**:
>   The potential application of our method in reinforcement learning (RL) contexts has been highlighted in the **Conclusion** section on page 10.
>
> - **Robustness to Challenging Scenarios**:
>   To demonstrate the robustness of our approach in poor data and out-of-distribution scenarios, we limited the test dataset for truck dynamics to include only winter test data. This data comprises aggressive maneuvers absent in the training dataset and out-of-distribution scenarios, such as braking in a μ-split scenario. These additions are detailed in the **Experimental Setup** section (pages 7–10) and in **Appendix B (Truck and Trailer Data Collected)** on page 17.
>
> ---
>
> ## Additional Enhancements:
>
> - **Expanded Baselines**:
>   We included comparisons with additional baselines, such as Koopman theory methods (e.g., Neural ODEs) and uncertainty quantification approaches (e.g., Bayesian NN and MC Dropouts). These results are detailed in the **Experimental Setup** section and **Appendix D (Additional Results)**.
>
> - **Visualizations**:
>   We added visualizations of loss functions across various baselines and environments, presented in the main experimental section and **Appendix D**.
>
> - **Runtime Analysis**:
>   A runtime analysis comparing our model to other baselines is now included in **Appendix D**.
>
> - **Real Vehicle Data Collection**:
>   **Appendix B** now includes an in-depth explanation of the data acquisition and processing methods used in the study.
>
> - **Motion Planning Details**:
>   Additional implementation details related to motion planning are provided in **Appendix D**.
>
> - **Implementation Details**:
>   **Appendix F** now includes a comprehensive description of the implementation specifics for our method and the baseline models.
>
> ---
>
> ## Supplementary Materials:
>
> - A revised manuscript with all updates clearly highlighted in blue.
> - Open-source code that encompasses the data collection process and model training procedures for the simulated environments.
>
> We thank the reviewer for the thoughtful feedback. We hope the revisions and newly incorporated materials, alongside our responses, sufficiently address the reviewer’s concerns and provide a clearer understanding of the potential and robustness of our proposed method.
>
> We hope these updates, along with our prior responses, encourage the reviewer to re-evaluate our submission positively. We remain fully open to any additional feedback or questions the reviewer may have.

---

### Official Review · Reviewer_EWkY · 2024-11-04

**Soundness:** 3
**Presentation:** 4
**Contribution:** 3
**Rating:** 8
**Confidence:** 4

**Summary:**

The authors propose to learn a dynamics prediction model that can adapt to different dynamical environment parameters for autonomous truck. The model leverages a transformer-based encoder of state and actions, plus Koopman-operator-based Bayesian learning for online adaptation. The method demonstrates SOTA performance compared to previous Koopman-operator-based approaches.

**Strengths:**

1. In 3.3 the effort to make the algorithm real-time in motion planning by using a variational encoder for action encoding such that sampling can be directly drawn from gaussian normal distribution is interesting and novel.
2. The writing of the paper is clear and easy to follow.
3. Combining Koopman operator with transformer-based encoding and adaptive control with Bayesian learning is an interesting paradigm.

**Weaknesses:**

1. More baselines outside of Koopman-based methods may be desired to connect the paper with other adaptive control and dynamics model learning paper, including but not limited to, models like neural ODE, PINN, or other uncertainty-aware approaches such as MC dropout.
2. No ablation study presents in the paper. See comments below.

Update in rebuttal
1. This weekness has been addressed from additional baselines that shows the author's proposed method is stronger.
2. The author now gives ablation study, but the results sometimes show only marginal performance improvement from the ablated components, which I assume is partially due to the simplicity of the environments. I encourage the authors to try on more challenging and long-horizon-demanding environments to better distinguish the effectiveness of the ablated component.

**Questions:**

1. (addressed in rebuttal) I assume the truck dataset would be heavily biased towards data of the vehicle driving straight with almost constant velocity, which may affect the quality of the model. If there are any effort to combat dataset imbalance, it would be beneficial to discuss.
2. (addressed in rebuttal) I’m not sure if the use of variational encoder to simplify online sampling is completely novel, but it would certainly be if this is the authors’ original idea. Otherwise maybe more related literature discussion is needed. I’ll leave this part to be answered by the authors and fellow reviewers during rebuttal phase.
3. (addressed in rebuttal, same comment as the weeknesses above) Some ablation studies may be needed. For example, BLAK without adaptation/bayesian learning. I’d like to know how much of the performance of the proposed method comes from the transformer + Koopman, versus how much is from bayesian learning.
4. (addressed in rebuttal) Additionally the authors could compare design choices with recent paper on transformer for adaptive vehicle dynamics prediction/control, such as details of how state/action are tokenized, encoding/decoding details, etc. Such as https://arxiv.org/abs/2409.15783, and https://arxiv.org/pdf/2310.08674

Other details
1. (addressed in rebuttal) Figure 1 caption is hard to follow. Suggest putting reference symbols (step A, B, C, etc.) on the plot.
2. (addressed in rebuttal) The authors start to use “BLAK” to refer to their method rather late into the paper (line 466) without first introducing what the term means.

---

> ### Author Response · Authors · 2024-11-24
> **Response to The Reviewer's Comments on Weaknesses**
>
> We thank the reviewer for the thoughtful and constructive feedback. We address each concern below.
>
>
> > **Weakness 1:** *More baselines outside of Koopman-based methods may be desired to connect the paper with other adaptive control and dynamics model learning papers, including but not limited to, models like neural ODE, PINN, or other uncertainty-aware approaches such as MC dropout.*
>
> **Answer:** We acknowledge the importance of comparing our method to a broader set of baselines beyond Koopman-based approaches. In the revised version of the manuscript, we will include detailed comparisons with Neural ODEs and MC Dropout techniques. We believe that this addition will highlight the robustness of our approach and establish a stronger connection to other adaptive control and dynamics learning paradigms.
>
> ---
>
> > **Weakness 2:** *No ablation study presents in the paper. See comments below.*
>
> **Answer:** We agree that ablation studies are crucial to understanding the contributions of each component of our method. Thus, we are working on an extensive ablation study and it will be added in the appendix of the revised manuscript. Specifically, the study will cover:
> 1. **Performance analysis:** We compare the performance of our model with and without Bayesian learning for several environments to show the impact of Bayesian learning in our approach.
> 2. **Trajectory Encoding:** We assess the benefit of trajectory encodings by experimenting with different trajectories length to validate the benefit of the approach.
>
> We believe these evaluations will provide a comprehensive understanding of how each element contributes to the overall performance and efficiency of our method, and address the reviewer’s concern. If there are any further studies that the reviewer thinks are useful, we would be more than happy to discuss them.

---

> > ### Author Response · Authors · 2024-11-24
> > **Additional Info**
> >
> > Finally, we will include visualizations of loss errors and open loop predictions, along with associated uncertainty bounds. We will add an implementation details section in the appendix. We hope our response adequately addresses the reviewer’s concerns, and we sincerely thank the reviewer for the positive and constructive feedback. We would be more than happy to answer any further questions the reviewer may have.

---

> ### Author Response · Authors · 2024-11-24
> **Response to the Reviewer's Questions 1/2**
>
> We appreciate the insightful questions and address them as follows:
>
> > **Question 1:** *I assume the truck dataset would be heavily biased towards data of the vehicle driving straight with almost constant velocity, which may affect the quality of the model. If there are any effort to combat dataset imbalance, it would be beneficial to discuss.*
>
>
> **Answer:** Indeed, data imbalance in the dataset poses a significant challenge for modeling truck dynamics. To address this, the test dataset, and especially the data collected during the winter test, only focused on trajectories that emphasize varied conditions, such as different road slopes, turns, and the mu-split scenario (where the right side of the truck is on dry asphalt and the left side is on ice.) These additional tests, especially the mu-split scenario, were conducted on specialized tracks in northern Sweden to ensure thorough coverage of challenging conditions. To give readers a deeper understanding of the difficulty of these scenarios and to provide a more rigorous evaluation of challenging maneuvers and distribution shifts, we limit the test set to include only these conditions. Moreover, we report test results for other maneuvers in the appendix. For the training set, straightforward driving data was deliberately under-sampled to constitute only 40% of the dataset. We have updated the manuscript to clarify these points.
>
> ---
>
> > **Question 2:** *I’m not sure if the use of variational encoder to simplify online sampling is completely novel, but it would certainly be if this is the authors’ original idea. Otherwise maybe more related literature discussion is needed. I’ll leave this part to be answered by the authors and fellow reviewers during rebuttal phase.*
>
> **Answer:** We confirm that the use of a variational autoencoder for dynamically feasible path sampling in real-time planning is indeed novel, to the best of our knowledge. We have updated the manuscript to clarify this.
>
> ---
>
> > **Question 3:** *Some ablation studies may be needed. For example, BLAK without adaptation/Bayesian learning. I’d like to know how much of the performance of the proposed method comes from the transformer + Koopman, versus how much is from Bayesian learning.*
>
> **Answer:** We agree with the reviewer that ablation studies are crucial for understanding the significance of each component. To address this, we will include an analysis of our method both with and without Bayesian learning to evaluate its impact. Furthermore, we will perform studies to assess the benefit of the trajectory encoding and report it in the ablation studies as well.
>
> ---

---

> ### Author Response · Authors · 2024-11-24
> **Response to the Reviewer's Questions 2/2**
>
> > **Question 4:** *Additionally the authors could compare design choices with recent paper on transformer for adaptive vehicle dynamics prediction/control, such as details of how state/action are tokenized, encoding/decoding details, etc. Such as https://arxiv.org/abs/2409.15783, and https://arxiv.org/pdf/2310.08674.*
>
> **Detailed answer:** We have incorporated these references in our related work section. As for the key design distinctions, they can be summed up as follows:
> 1. **AnyCar Paper:** This work employs a transformer-based model to predict vehicle dynamics. The state is represented as a 6-dimensional vector (x, y positions, heading angle, x, y velocities, and angular velocity), while the 2-dimensional action space includes throttle and steering angle. States and actions from historical sequences are linearly projected into 64-dimensional latent spaces via separate encoders, interleaved, and fed into the transformer with added positional encodings. The output is down-projected to predict future states.
> 2. **Adaptive Transformer Paper:** The paper proposes a model-based reinforcement learning framework using a System Identification Transformer (SIT) and an Adaptive Dynamics Model (ADM) for autonomous off-road driving. The state vector includes position, heading, linear, and angular velocities, while the action vector consists of throttle and steering angle. SIT uses a transformer-based architecture without positional encodings to extract a latent context vector summarizing past state-transition observations, enabling effective in-context adaptation. The ADM models state-transition distributions conditioned on the current state, action, and context vector, utilizing an LSTM followed by a fully connected layer to output a Gaussian distribution. For decision-making, the approach employs a Risk-Aware Model Predictive Path Integral (RA-MPPI) controller, ensuring safe and adaptive control under uncertainty.
> 3. **Our Approach:** We focus on motion planning rather than direct control. Our model utilizes a compact transformer architecture with a 32-dimensional embedding space. The system state is a 7-dimensional vector, encompassing longitudinal and lateral velocities and accelerations, the tractor's yaw angle, the trailer-to-tractor angle, and the front wheel slip angle. Actions include brake, thrust, and steering inputs. We use a trajectory encoder to transform historical state-action pairs into Koopman embeddings and an action encoder to produce future action embeddings. These embeddings are combined with the Koopman operator for future state prediction. Furthermore, our variational action encoder maps action embeddings to a standard Gaussian latent space, enabling efficient sampling for real-time motion planning. We will include detailed implementation parameters in the appendix for full transparency.
>
> We hope that this answer addresses the reviewer’s question. If there is anything that requires further elaboration, we’ll be more than happy to provide more details.

---

> ### Author Response · Authors · 2024-11-24
> **Response to Other Details**
>
> > **Detail 1:** *Figure 1 caption is hard to follow. Suggest putting reference symbols (step A, B, C, etc.) on the plot.*
>
> **Answer:** We appreciate the reviewer’s attention to these details. We have added reference symbols to the plot and made it easier to follow. Thank you for suggesting this improvement.
>
> ---
>
> > **Detail 2:** *The authors start to use “BLAK” to refer to their method rather late into the paper (line 466) without first introducing what the term means.*
>
> **Answer:** We have now introduced the term “BLAK” earlier in the text and provided the proper explanation of what it stands for.

---

> ### Author Response · Authors · 2024-11-30
> **Key Revisions and Enhancements**
>
> We sincerely thank the reviewer for their insightful comments and constructive feedback. Below, we outline the key revisions made in response to the reviewer’s suggestions:
>
> ## Key Revisions Addressing the Reviewer's Concerns:
>
> 1. **Comparative Analysis**:  As recommended, we have incorporated comparisons with Bayesian neural networks, MC Dropouts, and Neural ODEs across all six environments, including truck dynamics and other simulated scenarios. These results are detailed in the **“Experimental Results”** section (end of page 9) for the truck dynamics dataset and further elaborated in **Appendix D, “Additional Results,”** on pages 20–23 for simulated environments.
>
> 2. **Ablation Studies**:  We conducted ablation studies to evaluate the advantages of Bayesian learning and the impact of historical trajectory length on performance. These findings are now included in **Appendix E** on page 26 of the revised manuscript.
>
> 3. **Writing Improvements**:  The **Introduction** and **Background & Related Work** sections (pages 2–3) now better emphasize the novelties and contributions of our method. Additionally, **Figure 1** has been refined to include the suggested improvements, and the term **"BLAK"** has been introduced in the abstract for clarity.
>
> ---
>
> ## Additional Enhancements:
>
> - **Implementation Details**:  Appendix F now provides detailed descriptions of the implementation for both our proposed method and baseline models.
>
> - **Real Vehicle Data Collection**:  Appendix B includes a comprehensive explanation of the data acquisition and processing methods used in the study.
>
> - **Visualizations**:  Visualizations of loss functions across different baselines and environments have been added, appearing in both the main experimental section and **Appendix D**.
>
> - **Runtime Analysis**:  A runtime comparison between our model and other baselines is presented in **Appendix D**.
>
> - **Motion Planning Details**:  Additional implementation details for motion planning have been included in **Appendix D**.
>
>
> ---
>
> ## Supplementary Materials:
>
> - A revised manuscript with all changes highlighted in blue for clarity.
> - Open-source code, including the data collection process and model training procedures.
>
> We are deeply grateful for the reviewer’s thoughtful suggestions and remain available to address any further questions or feedback.

---

> > ### Comment · Reviewer_EWkY · 2024-11-30
> > **Good Paper; Encourage Better Ablation in Final Submission**
> >
> > I thank the authors for their thorough rebuttal. I've updated my scores accordingly and recommend accepting the paper.
> >
> > I've edited the original review to indicate issues that have been resolved, although I left additional comments on more distinguishing ablation studies.

---

> > > ### Author Response · Authors · 2024-12-02
> > >
> > > We sincerely value the reviewer’s thoughtful and constructive feedback, as well as their positive evaluation of our work. In the next version of our manuscript, we will ensure the inclusion of additional ablation studies. Once again, we thank the reviewer for their encouraging remarks and remain fully open to any further questions or insights they may have.

---

### Author Response · Authors · 2024-11-24
**General Comment for All Reviewers**

We sincerely thank all the reviewers for their thoughtful feedback and valuable insights. In response to the reviewers' suggestions, we are currently conducting experiments with new baseline models. A revised manuscript incorporating these updates will be submitted shortly.

We are open to addressing any further comments or questions from the reviewers

---

### Author Response · Authors · 2024-11-30
**General Comment to Reviewers: Summary of Revisions and Enhancements**

We sincerely thank all reviewers for their valuable and constructive feedback. In response to your insightful comments, we have implemented the following key updates and enhancements to address your concerns and suggestions:

---

## Key Revisions and Clarifications:

- **Clarification of Contributions**:
  We have explicitly clarified how our method differs from existing work in the literature, emphasizing its novelty and unique contributions. These updates are detailed in the **Introduction** and **Related Work** sections.

- **Theoretical and Algorithmic Justifications**:
  The theoretical foundations and algorithmic contributions of our method have been further elaborated. This includes a detailed justification of using the transformer encoder (page 4) and expanded proofs with clarified assumptions (**Appendix A**).

- **Enhanced Figures and Visualizations**:
  Figures have been improved to better outline the steps of our algorithm and distinguish between key elements.

---

## Enhanced Evaluation and Testing:

- **Expanded Baselines**:
  We incorporated comparisons with three additional baselines, such as non Koopman-based methods (Neural ODEs) and uncertainty quantification techniques (Bayesian NN and MC Dropouts). These results are presented in the **Experimental Setup** section and **Appendix F**.

- **Robustness Testing**:
  To demonstrate the robustness of our method, we tested it on challenging, out-of-distribution scenarios, such as winter test data containing aggressive maneuvers and μ-split braking. These experiments are detailed in the **Experimental Setup** section (pages 7–10) and **Appendix B** (page 17).

- **Runtime Analysis**:
  A runtime analysis comparing our method to other baselines has been added in **Appendix D**.

- **Loss Visualizations**:
  Visualizations of loss functions across various baselines and environments have been added in both the main manuscript and **Appendix D**.

- **Ablation Studies**:
  Ablation studies were conducted to evaluate the benefits of Bayesian learning and analyze the impact of sequence length on performance. These results are presented for varying sequence lengths in **Appendix E**.

---

## Additional Enhancements:

- **Motion Planning and Computational Evaluation**:
  We added a dedicated section on motion planning with a pseudo-code algorithm and included a computational assessment in **Appendix D** (pages 23–25).

- **Data Collection Details**:
  **Appendix B** now includes a detailed explanation of the data acquisition and processing methods used in both real-world and simulated environments.

- **Improved Implementation Details**:
  Comprehensive implementation details, including parameter values for our method and baselines, are now provided in **Appendix F** (pages 27–28).

---

## Supplementary Materials:

- **Revised Manuscript**:
  A revised manuscript with all updates clearly highlighted in blue for transparency.

- **Open-Source Code**:
  Open-source code, including the data collection process and model training procedures for simulated environments.

---

We deeply value the time and effort each reviewer has invested in providing their thoughtful feedback. We believe the revisions and additional materials comprehensively address the concerns raised and provide a more robust and transparent evaluation of our proposed method. We remain open to any additional questions or suggestions the reviewers may have and would be more than happy to further engage in meaningful discussions.

---

### Author Response · Authors · 2024-12-02
**Available for Last-Minute Questions**

As the author-reviewer discussion period draws to a close, we extend our gratitude to the reviewers for their valuable feedback and insights.

We are fully available to promptly address any remaining questions the reviewers may have during these final hours.

---

### Meta-Review · Area_Chair_M6Hg · 2024-12-17

**Metareview:**

This submission contributes fundamentally interesting and new techniques related to change detection for dynamical systems, with an interesting application in autonomous trucks. However, the reviewers have several technical concerns regarding consistency and clarity of writing, as well as subject matter scope. For these reasons, it is below the bar for acceptance at this time.

**Additional Comments On Reviewer Discussion:**

not applicable.

---

### Decision · Program_Chairs · 2025-01-22

Reject